EMBO
Molecular Medicine

# Reprogramming of profibrotic macrophages for treatment of bleomycin-induced pulmonary fibrosis

Fenghua Zhang[1] (ID), Ehab A Ayaub[2], Bingbing Wang[1], Estela Puchulu-Campanella[1] (ID), Yen-Hsing Li[1], Suraj U Hettiarachchi[1], Spencer D Lindeman[1], Qian Luo[1], Sasmita Rout[1], Madduri Srinivasarao[1], Abigail Cox[3], Konstantin Tsoyi[2], Cheryl Nickerson-Nutter[4], Ivan O Rosas[2] & Philip S Low[1,*] (ID)

## Abstract

Fibrotic diseases cause organ failure that lead to ~45% of all deaths in the United States. Activated macrophages stimulate fibrosis by secreting cytokines that induce fibroblasts to synthesize collagen and extracellular matrix proteins. Although suppression of macrophage-derived cytokine production can halt progression of fibrosis, therapeutic agents that prevent release of these cytokines (e.g., TLR7 agonists) have proven too toxic to administer systemically. Based on the expression of folate receptor β solely on activated myeloid cells, we have created a folate-targeted TLR7 agonist (FA-TLR7-54) that selectively accumulates in profibrotic macrophages and suppresses fibrosis-inducing cytokine production. We demonstrate that FA-TLR7-54 reprograms M2-like fibrosis-inducing macrophages into fibrosis-suppressing macrophages, resulting in dramatic declines in profibrotic cytokine release, hydroxyproline biosynthesis, and collagen deposition, with concomitant increases in alveolar airspaces. Although nontargeted TLR7-54 is lethal at fibrosis-suppressing doses, FA-TLR7-54 halts fibrosis without evidence of toxicity. Taken together, FA-TLR7-54 is shown to constitute a novel and potent approach for treating fibrosis without causing dose-limiting systemic toxicities.

**Keywords** bleomycin; folate receptor β; idiopathic pulmonary fibrosis; macrophages; toll-like receptor 7

**Subject Categories** Pharmacology & Drug Discovery; Respiratory System

## Introduction

Fibrotic diseases, in which normal tissue is replaced by scar tissue leading ultimately to organ failure, are reported to be responsible for ~45% of all deaths in the United States (Nanchahal & Hinz, 2016). Idiopathic pulmonary fibrosis (IPF) is an interstitial lung disease that results from excessive deposition of collagen, leading to progressive stiffening of the lung and the consequent loss of ability to mediate gas exchange (Plantier *et al*, 2018). Due to the constitutive decline in vital capacity, median survival following diagnosis is estimated at only 2–3 years (Warheit-Niemi *et al*, 2019), with the severity of associated morbidities (e.g., chronic hypoxia, fatigue, muscle and joint pain, persistent coughing, and loss of mobility) increasing continuously during later stages of the disease. With ~40,000 new cases of IPF diagnosed per year in the United States and most cases ending in death, public awareness of the pathology is finally increasing (https://ghr.nlm.nih.gov/condition/idiopathic-pulmonary-fibrosis#statistics).

Despite the seriousness of IPF, few options have been developed for the treatment of the disease, with nearly all therapies designed to mitigate symptoms or retard progression, but none capable of curing the pathology (Somogyi *et al*, 2019). Thus, oxygen therapy can improve comfort and lifestyle, but has little effect on disease progression (Graney *et al*, 2017). Although two FDA-approved drugs, pirfenidone and nintedanib, may slow advancement of the disease, neither can reverse existing fibrosis nor halt production of further fibrosis (Dempsey *et al*, 2019). Given the high probability of mortality associated with IPF, there is clearly a major need for new strategies to halt or even reverse the disease.

Two major cell types appear to play essential roles in the development of IPF (Prasse *et al*, 2006; Byrne *et al*, 2015; Pakshir & Hinz, 2018). First, lung fibroblasts that derive from peripheral blood fibrocytes (Darby *et al*, 2014), alveolar epithelial cells that undergo epithelial-to-mesenchymal transition (EMT) (Kage & Borok, 2012), or endogenous lung fibroblasts become activated to form myofibroblasts that in turn secrete collagen and other extracellular matrix proteins that stiffen the lung (Habiel & Hogaboam, 2017). Second, activated macrophages, which derive from tissue-resident macrophages or peripheral blood monocytes, induce activation of the aforementioned fibroblasts via secretion of CCL18, TGFβ, IL-1β, and/or PDGF, promoting their secretion of collagen (Prasse *et al*, 2006; Wilson *et al*, 2010; Braga *et al*, 2015; Byrne *et al*, 2016; Wynn & Vannella, 2016; Misharin *et al*, 2017). In later stages of IPF, the activated macrophages and

1 Department of Chemistry and Institute for Drug Discovery, Purdue University, West Lafayette, IN, USA
2 Division of Pulmonary and Critical Care Medicine, Brigham and Women's Hospital, Harvard Medical School, Boston, MA, USA
3 Department of Comparative Pathobiology, Purdue College of Veterinary Medicine, West Lafayette, IN, USA
4 Three Lakes Partners, Northbrook, IL, USA
*Corresponding author. Tel: +1-765-494-5273; E-mail: plow@purdue.edu

myofibroblasts are thought to even cross-stimulate each other, resulting in a vicious cycle that assures propagation of fibrosis throughout the lung (Prasse *et al*, 2006; Pakshir & Hinz, 2018).

In the study below, we test the hypothesis that reprogramming of profibrotic to anti-fibrotic macrophages can alleviate many characteristics of pulmonary fibrosis. Based on data demonstrating that IPF macrophages might be disproportionately biased toward an anti-inflammatory (M2-like) tissue healing/repair phenotype (Prasse *et al*, 2006; Braga *et al*, 2015; Byrne *et al*, 2015), we screened for immune modulators that might convert IPF macrophages into an anti-fibrotic state. We demonstrate below that targeting of a potent toll-like receptor 7 (TLR7) agonist specifically to IPF macrophages can successfully reprogram these cells, causing a significant decrease in markers of fibrosis (i.e., CCL18, hydroxyproline, and collagen), a concomitant increase in hallmarks of anti-fibrotic activity (i.e., CXCL10, IFNα, IFNγ, and CD86) (Diaz & Jiménez, 1997; Jiang *et al*, 2010; Lutherer *et al*, 2011; Braga *et al*, 2015; Murray, 2017), and the anticipated expansion of alveolar airspaces.

## Results

### Analysis of profibrotic macrophage reprogramming *in vitro*

As hypothesized in the introduction, one strategy for halting progression of IPF might be to selectively reprogram the lung macrophages from a profibrotic M2-like phenotype to an anti-fibrotic M1-like phenotype. In search for a cell surface receptor on profibrotic lung macrophages for use in targeted delivery of anti-fibrotic drugs, we noted that folate receptor β (FRβ) is highly expressed on macrophages from both human IPF lungs (Hu *et al*, 2019) and murine bleomycin (BLM)-induced fibrotic lungs (Nagai *et al*, 2010), but largely absent from cells in normal healthy tissues (Low *et al*, 2007) (see also Fig EV1). Since we (Low & Kularatne, 2009) and others (Sudimack & Lee, 2000) had already developed methods to target drugs to folate receptor-expressing cells, we elected to explore the use of these same targeting methods to deliver anti-fibrotic drugs to FRβ$^+$ macrophages.

Although reports from other laboratories had demonstrated that reprogramming of M2-like to M1-like macrophages could be induced by TLR7 agonists (Rodell *et al*, 2018; Mullins *et al*, 2019), toxicities associated with systemic administration of TLR7 agonists had precluded use of such drugs for the treatment of IPF (Savage *et al*, 1996; Harrison *et al*, 2004; Geller *et al*, 2010; Biffen *et al*, 2012). We therefore hypothesized that if we could target a TLR7 agonist selectively to IPF lung macrophages, we might be able to selectively reprogram the IPF macrophages without causing any systemic toxicity. Based on the aforementioned observations that folate can be exploited to deliver attached drugs to folate receptor-expressing cells, we decided to investigate whether folate might be used to target TLR7 agonists to IPF lung macrophages.

As an initial test of this strategy, we examined whether THP-1 cells (a human monocytic cell line that can be induced to adopt an M2-like phenotype and produce profibrotic cytokines upon stimulation with IL-4, IL-6 plus IL-13) (Fernando *et al*, 2014; Genin *et al*, 2015) might be reprogrammed to an anti-fibrotic phenotype upon treatment with a nontargeted TLR7 agonist *in vitro*. For this purpose, we induced THP-1 cells with IL-4, IL-13 plus IL-6 and examined the mRNA levels of both an M2 polarization marker (CD206) and two classic profibrotic cytokines (CCL18, IL-1β) (Prasse *et al*, 2006; Wilson *et al*, 2010) following incubation in the presence and absence of a potent TLR7 agonist (i.e., TLR7-54; ref. Shukla *et al*, 2010; see Appendix Fig S2A). As shown in Fig 1A, incubation with nontargeted TLR7-54 induced a decrease in all three M2/profibrotic markers (i.e., CCL18, IL-1β, and CD206), suggesting that the TLR7 agonist can indeed promote a shift in profibrotically polarized THP-1 cells toward a less fibrotic phenotype. Because studies below explore the ability of the same TLR7 agonist to reprogram murine profibrotic macrophages to an anti-fibrotic phenotype, we also examined the effect of TLR7-54 on the phenotype of IL-4, IL-13 with or without IL-6-stimulated murine bone marrow-derived macrophages. As shown in Appendix Fig S3, TLR7 agonist can similarly reprogram murine macrophages. To characterize the specificity of TLR7-54, we next incubated TLR7-54 for 6 h with either THP-1-NF-κB-luc cells or the same cells transduced to express human TLR7. As shown in Fig EV2, TLR7-54 failed to activate NF-κB in the former cell line (i.e., consistent with ref. Eng *et al*, 2018), but strongly induced NF-κB activation in the latter. These data demonstrate that activation of the monocytic THP-1 cell line upon treatment with TLR7-54 requires expression of TLR7.

To determine whether a folate receptor-targeted form of the same TLR7 agonist might enable a similar activation of FRβ-expressing THP-1-induced macrophages (see Appendix Fig S4), we then prepared a folate-conjugated version of TLR7-54 (FA-TLR7-54; Appendix Figs S1 and S2A) in which a linker connecting folate to TLR7-54 was constructed with a disulfide bond that would allow release of unmodified TLR7-54 following its internalization into the reducing environment of intracellular endosomes (Yang *et al*, 2006). As shown in Fig 1A (gray bars), examination of the same profibrotic markers following treatment with FA-TLR7-54 yielded similar changes to those seen with nontargeted TLR7-54, only the impact of the targeted drug was somewhat less because the targeted drug was designed to only enter cells by folate receptor-mediated endocytosis (Varghese *et al*, 2014).

Finally, to ensure that the above mRNA analyses accurately reflect the levels of profibrotic cytokines, the concentrations of CCL18 and IL-1β polypeptides in the THP-1 supernatants were quantitated by ELISA. As shown in Fig 1B, both nontargeted TLR7-54 and folate-targeted FA-TLR7-54 were found to induce reductions in CCL18 and IL-1β when incubated continuously with agonist for 48 h. More importantly, when drug exposure was limited to only 2 h (i.e., to mimic the exposure time of rapidly excreted folate conjugates *in vivo*; ref. Srinivasarao *et al*, 2015), FA-TLR7-54 was found to be superior (Fig 1C), likely because it is captured and internalized by the folate receptors, i.e., preventing its removal during exchange of culture media after 2 h.

Next, to confirm the ability of TLR7 agonists to reverse the profibrotic properties of macrophages in a more physiologically relevant model, we isolated monocytes from human peripheral blood and differentiated them into M2-like macrophages before exposing them to TLR7-54 or FA-TLR7-54. As shown in Fig 2A–D, FA-TLR7-54 was able to suppress both mRNA levels of the profibrotic markers (Arg1, CD206, and CD163) and production of the profibrotic cytokine, CCL18. Moreover, as revealed in panels 2E and 2F, FA-TLR7-54 was able to elevate production of the anti-fibrotic cytokines, CXCL10 and IL-6, in these same M2-like macrophages. More importantly, shifts

in both sets of cytokine expression were inhibited by blockade of unoccupied folate receptors with excess folate-glucosamine (FA-glucosamine, i.e., a competitive inhibitor of FRβ-binding) (Gent et al, 2013), confirming that uptake of FA-TLR7-54 requires an unoccupied folate receptor. Because human IPF lungs are also comprised predominantly of FRβ$^+$ M2-polarized macrophages that are thought to contribute most prominently to development of fibrosis (Prasse et al, 2006; Braga et al, 2015; Byrne et al, 2015), these data suggest that FA-TLR7-54 might similarly reprogram human IPF macrophages to a less profibrotic phenotype.

**Analysis of profibrotic macrophage reprogramming in vivo**

Encouraged by these results, we next undertook to determine whether macrophages in pulmonary fibrotic lungs might be specifically targeted with folate-linked drugs. After testing multiple methods for induction of pulmonary fibrosis in mice, we selected a protocol where 0.75 mg/kg BLM is instilled into the lungs of C57BL/6 mice and the mice are allowed to progress through both inflammatory and fibrotic stages of fibrosis prior to initiation of therapy. Although inadequacies still exist, this methodology has been described in the Official American Thoracic Society Workshop Report as "the best-characterized animal model available for preclinical testing" of IPF (Jenkins et al, 2017; Tashiro et al, 2017). As shown in Fig EV1, mice treated using this protocol display indications of fibrosis by day 7 post-BLM treatment and this nascent fibrosis develops into severe fibrosis by day 14. Progression of the pathology then continues for an additional 2–5 days before it begins to spontaneously resolve by day 21.

To evaluate whether profibrotic lung macrophages in these mice can be targeted with folate-linked drugs, we injected a folate-linked

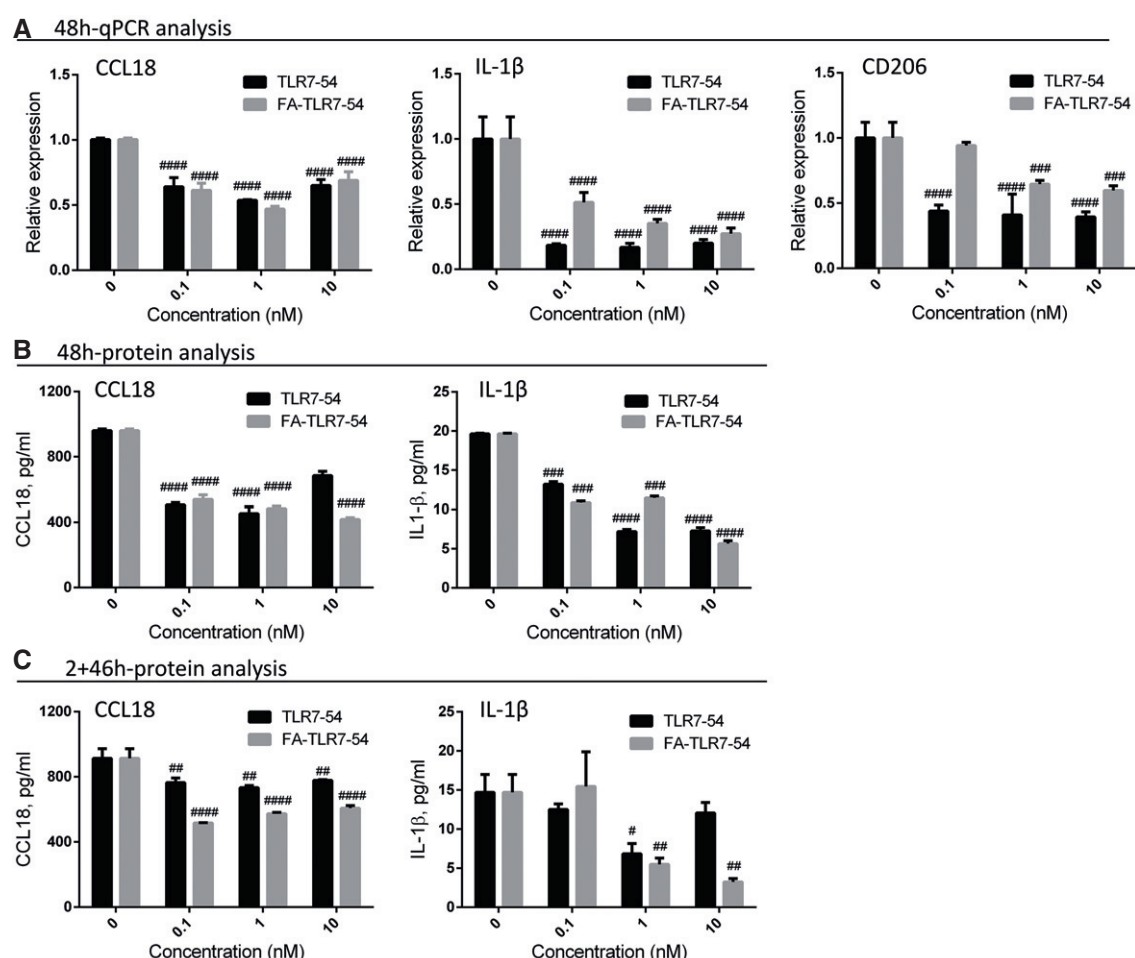

**Figure 1. Nontargeted TLR7 agonist (TLR7-54) and folate-targeted TLR7 agonist (FA-TLR7-54) downregulate profibrotic macrophage markers.**

A–C  Human monocytic (THP-1) cells were induced to acquire an M2-like phenotype (see Materials and Methods) and treated with different concentrations of TLR7-54 or FA-TLR7-54 for either 48 h (A–B) or 2 h (C). In the latter case, after the 2 h incubation, culture medium was replaced with drug-free medium and incubation was continued for 46 h. All treatment groups were then analyzed by qPCR for gene expression, and cell supernatants were analyzed for secreted cytokines by ELISA. (A) Changes in mRNA levels of indicated profibrotic macrophage markers by induced by different concentrations of TLR7-54 and FA-TLR7-54 (n = 3, technical replicates). (B-C) Changes in CCL18 and IL-1β in the culture media induced upon treatment with TLR7-54 and FA-TLR7-54 for the treatment regimens (n = 3, technical replicates).

Data information: Mean ± SD. Statistical significance between TLR7-54- or FA-TLR7-54-treated groups versus M2-untreated group compared using Dunnett's multiple comparison test ($^#P < 0.05$, $^{##}P < 0.01$ $^{###}P < 0.001$, $^{####}P < 0.0001$).

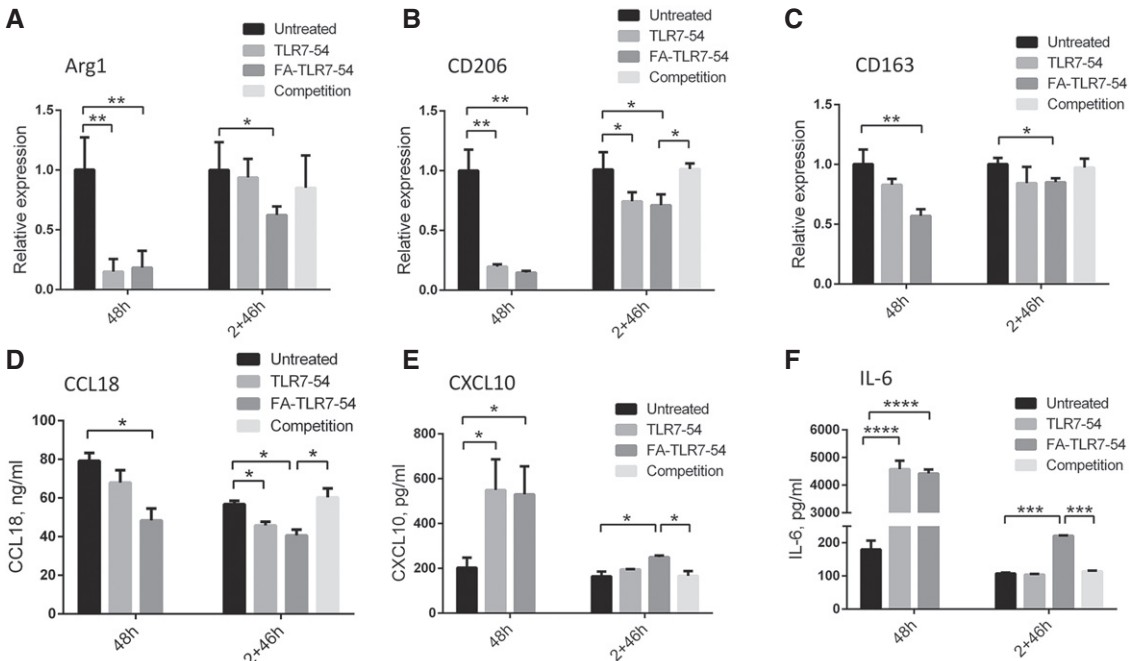

**Figure 2. Both targeted and nontargeted TLR7 agonists reprogram human monocyte-derived profibrotic macrophages to an anti-fibrotic phenotype.**

A–F  M2-induced human monocyte-derived macrophages were treated with 100 nM of the indicated drug either continuously for 48 h, or initially for 2 h in the presence or absence of FA-glucosamine (competition) followed by 46 h in the absence of drug (2+46 h), as described in Fig 1. mRNA levels of profibrotic markers, Arg1 (A), CD206 (B), and CD163 (C), and protein levels of secreted profibrotic CCL18 (D) and anti-fibrotic cytokines, CXCL10 (E) and IL-6 (F) (n = 3, technical replicates), were then determined. Changes in both sets of cytokines were inhibited by blockade of unoccupied folate receptors with excess FA-glucosamine (2+46 h, competition).

Data information: Mean ± SD. Statistical significance between groups was determined using unpaired two-tailed t-test (*P < 0.05, **P < 0.01, ***P < 0.001, ****P < 0.0001).

fluorescent dye (OTL38) (Mahalingam et al, 2018) into the tail veins of healthy or BLM-treated mice and examined uptake of the dye in the major organs. As shown in Fig 3A, OTL38 fluorescence was only observed in the kidneys of healthy mice (i.e., its major site of excretion), with little or no uptake in other tissues. In contrast, injecting OTL38 into BLM-treated mice yielded not only the fluorescence in the kidneys, but also pronounced accumulation in the fibrotic lungs. That this lung uptake was largely mediated by folate receptors could be demonstrated by the blockade of lung accumulation when the BLM-treated mice were simultaneously injected with 200-fold excess FA-glucosamine. These data demonstrate that a folate-targeted molecule will bind selectively to folate receptor-expressing cells in fibrotic lungs without accumulating significantly in any other tissues of the body.

To determine what cell type might be capturing the OTL38 in the lungs of BLM-treated mice, lungs from the above animals were digested with collagenase (see Materials and Methods) and examined by flow cytometry for cell-specific folate-dye uptake. As shown in Fig 3B–D, all three BLM-treated groups showed similar percentages of macrophages, and no macrophages displayed any fluorescence in mice not injected with OTL38. In contrast, ~20% of the macrophages from OTL38-injected fibrotic mice showed significant folate-targeted dye retention. That this folate-dye uptake was folate receptor-mediated could be demonstrated by the observation that concurrent tail vein injection of 200-fold excess FA-glucosamine blocked considerable folate-dye retention, demonstrating that

accumulation of the dye required unoccupied folate receptors. Moreover, when the pulmonary macrophage subpopulations were subsequently segregated by flow cytometry and examined for FRβ expression, FRβ was largely limited to monocyte-derived alveolar macrophages (Mono-AMs), with considerably less FRβ on the small population of interstitial macrophages and barely any expression on tissue-resident alveolar macrophages (Fig EV3). This localization of FRβ to Mono-AMs is informative, because these subpopulations are thought to be disproportionately responsible for inducing pulmonary fibrosis (Misharin et al, 2017). Significantly, this conclusion is also supported by data showing that FRβ expression is essentially nondetectable in healthy lungs (Fig 3A) but increases dramatically during development of fibrosis in BLM-treated lungs (Fig EV1, panels A and B). FRβ expression is also prominently expressed in lungs of human IPF patients (Fig EV1D), but not in lungs of healthy individuals (Fig EV1C).

With an ability to target drugs to fibrotic lung macrophages established, the question next arose whether a folate-targeted TLR7 agonist might be capable of reprogramming the profibrotic macrophages in fibrotic lungs. As an initial exploration of this question, BLM-treated mice were intravenously injected on day 10 with either vehicle (3% DMSO in PBS), 10 nmol TLR7-54 or FA-TLR7-54, and after 1 and/or 4 h bronchoalveolar lavage fluid (BALF) and lungs were isolated and analyzed. As seen in Fig 4A, mRNA levels of profibrotic markers Arg1, CD206, and CD163 were all suppressed in lung tissue-derived macrophages following a single dose of either

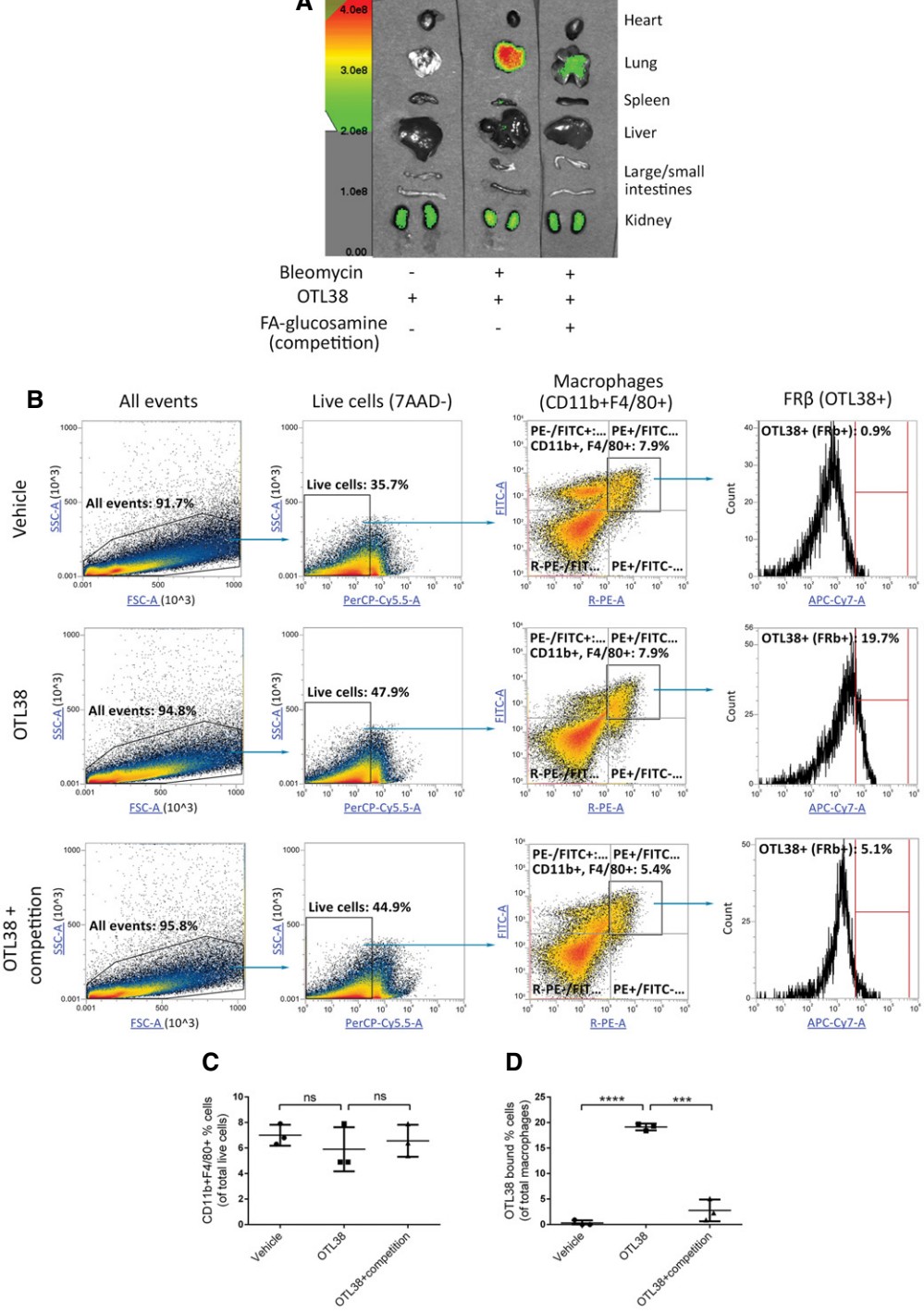

**Figure 3. Evaluation of folate-dye conjugate targeting to FRβ⁺ macrophages in lungs of mice with pulmonary fibrosis.**

A   Healthy mice or BLM-induced mice were tail vein injected on day 10 with 10 nmol OTL38 in the absence or presence of 200-fold excess FA-glucosamine to block all folate receptors. After 2 h, mice were euthanized, resected, and imaged for fluorescence intensity (*n* = 5).

B   Alternatively, lungs from mice injected with 100 nmol OTL38 in the presence or absence of 200-fold excess FA-glucosamine were collagenase digested and stained with 7-AAD plus antibodies to CD11b and F4/80 prior to FACS analysis (*n* = 3). Representative plots showing the gating strategy yielding live macrophages (7-AAD⁻ CD11b⁺ F4/80⁺) and OTL38-positive macrophages are shown.

C   Percentages of live macrophages in BLM-induced mice (*n* = 3).

D   Percentages of lung macrophages that accumulated OTL38 *in vivo* (*n* = 3).

Data information: Mean ± SD. Statistical significance between groups was compared using unpaired two-tailed *t*-test (***P* < 0.001, ****P* < 0.0001).
Source data are available online for this figure.

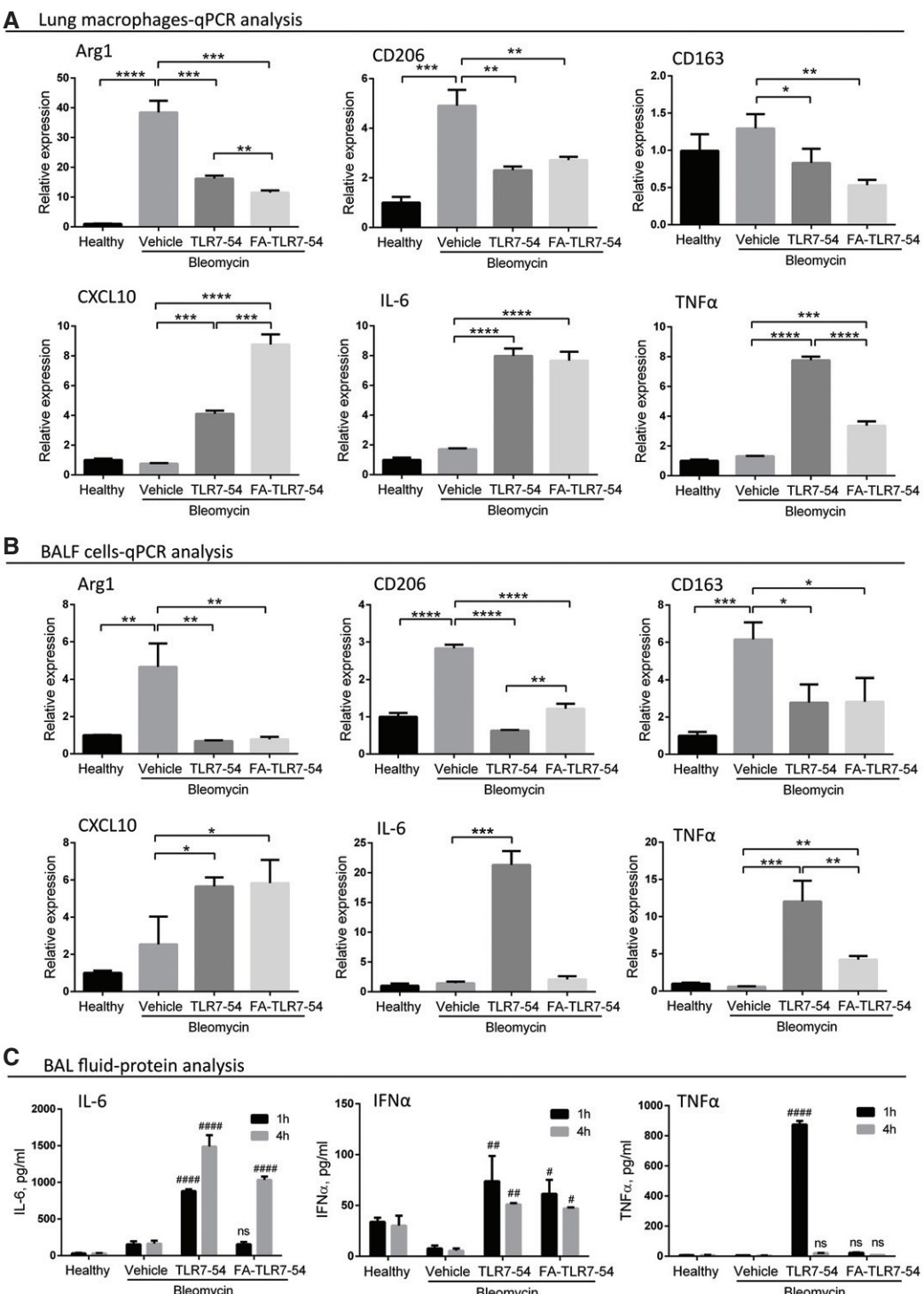

**Figure 4. Effect of a single dose of targeted or nontargeted TLR7-54 on phenotypic markers of pulmonary fibrosis macrophages *in vivo*.**

A–C Healthy mice or BLM-induced mice were injected intravenously on day 10 with either vehicle (3% DMSO in PBS), or 10 nmol TLR7-54 or FA-TLR7-54 dissolved in vehicle, and 1 or 4 h later sacrificed to collect both lungs and bronchoalveolar lavage fluid (BALF). (A) Lungs were digested, and macrophages were isolated by flow cytometry prior to analysis for expression of the indicated mRNAs by qPCR (n = 3). (B) BALF cells were pelleted and similarly analyzed for the indicated mRNAs (n = 3). (C) BALF supernatant was also collected and analyzed by ELISA for IL-6, IFNα, and TNFα (n = 3).

Data information: Mean ± SD. Statistical significance between groups was compared using unpaired two-tailed *t*-test (*P < 0.05, **P < 0.01, ***P < 0.001, ****P < 0.0001). Statistical significance between groups at 1 h or 4 h time point was compared using Dunnett's multiple comparison test (#P < 0.05, ##P < 0.01, ####P < 0.0001).

targeted or nontargeted TLR7 agonist, while mRNA levels of anti-fibrotic markers CXCL10, IL-6, and TNFα were elevated in the same macrophages.

To further document the ability of TLR7 agonist to reprogram fibrotic lung macrophages, BALF macrophages were isolated and similarly analyzed, and seen in Fig 4B, which yielded the same shift from profibrotic to anti-fibrotic markers. Because the expected rise in anti-fibrotic cytokine (protein) levels was also observed in the same BALF (Fig 4C), we conclude that TLR7 agonists can reprogram fibrotic lung macrophages to a more anti-fibrotic state *in vivo*. Interestingly, the FA-TLR7-54 did not induce the upregulation of TNFα as strongly as the nontargeted TLR7-54 did (Fig 4A–C).

Next, to determine whether a folate-targeted TLR7 agonist might be capable of suppressing fibrosis, BLM-treated mice were intravenously injected every other day beginning on day 10 with either vehicle or FA-TLR7-54 (because nontargeted TLR7-54 caused severe body weight loss followed by death (see Fig 5B), nontargeted TLR7-54 could not be similarly evaluated *in vivo*). Mice were then sacrificed on day 21 and subjected immediately to bronchoalveolar lavage followed by resection of lungs for immunohistochemistry and quantitation of collagen and hydroxyproline (Fig 5A). As shown in Fig 5C–E, quantitative real-time PCR (qPCR) analysis of profibrotic markers in the BALF macrophages revealed that Arg1 (Grasemann *et al*, 2015), MMP9 (Kui Tan *et al*, 2013), and TIMP3 (García-Alvarez *et al*, 2006) were all elevated in BLM-induced mice relative to controls. More importantly, parallel studies demonstrated that the same profibrotic markers were suppressed when treated with FA-TLR7-54, i.e., yielding levels of the fibrotic markers similar to those seen in healthy mice. Immunofluorescence staining of an M2 marker (CD206) in the fibrotic lungs further showed that the CD206-positive macrophages were abundant in the nontreated fibrotic lungs but strongly reduced in the FA-TLR7-54-treated lungs (Fig EV4). Consistent with these data, quantitation of anti-fibrotic markers revealed that transcripts of CD86 (qPCR) and concentrations of IFNγ (ELISA of BALF supernatant) were both elevated following treatment with FA-TLR7-54 (Fig 5F and H). Taken together with the observed upregulation of IRAK4 (i.e., a marker of TLR activation, Fig 5G, ref. Cushing *et al*, 2017), these data confirm that repeated administration of folate-targeted TLR7 agonist can reprogram fibrotic macrophages to anti-fibrotic phenotype in the lungs of BLM-treated mice.

To determine whether the above reprogramming of fibrotic lung macrophages might have impacted the BLM-induced development of fibrosis, lung tissue from the above mice was sectioned and stained with hematoxylin–eosin (H&E) and Masson's trichrome for the evaluation of tissue morphology and extracellular collagen deposition, respectively (Fig 5I). As shown in the H&E stains of panel I (top row), healthy lungs contained an abundance of air sacs surrounded by thin reticular membranes. In contrast, BLM-induced lungs displayed far fewer alveoli with pronounced deposition of extracellular matrix where air sacs once existed. Importantly, BLM-instilled mice treated beginning on day 10 with FA-TLR7-54 exhibited a lung architecture that resembles that of healthy mice (panel I), suggesting that targeting of TLR7-54 to fibrotic lung macrophages can suppress the major hallmarks of pulmonary fibrosis. That this prevention of fibrosis indeed involves blockade of collagen deposition is in fact documented by trichrome staining of parallel lung sections, where the blue collagen stain is seen to be strongly suppressed in mice treated with FA-TLR7-54 (panel I; middle row). Moreover, that the FA-TLR7-54 therapy also causes a reduction in the number of activated myofibroblasts is shown in panel I (lower row), where alpha-smooth muscle actin (α-SMA), a marker of activated myofibroblasts (Rao *et al*, 2014), is seen to decline following FA-TLR7-54 administration, i.e., consistent with the hypothesis that M2-like macrophages are prominent in generating the fibroblast-activating cytokines (Zhang *et al*, 2018). Finally, to further confirm that the targeted TLR7 agonist is indeed impacting production of collagen, hydroxyproline (a major component of collagen) (Fig 5J) and Ashcroft score (Fig 5K) were quantitated in the affected lungs. As shown in Fig 5J and K, induction of fibrosis induces a large increase in the hydroxyproline content and Ashcroft score, and this increase is suppressed upon treatment with FA-TLR7-54.

Finally, because use of a nontargeted TLR7 agonist to treat IPF in humans has been prevented by its systemic activation of the immune system, the question naturally arose whether any obvious toxicities might have accompanied systemic administration of FA-TLR7-54 in the mice. To address this question, BLM-induced mice were treated every other day beginning on day 10 with 0, 1, 3, or 10 nmol FA-TLR7-54 and body weight, lung hydroxyproline content, and histological analyses were performed on day 21. As seen in Fig 6A, no difference in weight loss was observed between mice treated with 0, 1, 3, or 10 nmol folate-targeted TLR7-54, suggesting that no gross toxicity was caused by repeated dosing of FA-TLR7-54. That these treatments were still exerting their anticipated effects on lung fibrosis could be seen from a comparison of total BALF cell numbers (panel B) and the hydroxyproline contents of the various lung hydrolyzates (panel C), which showed dose-dependent efficacy. More importantly, detailed analyses of the lung histology demonstrated that as the dose of FA-TLR7-54 increased, lung histology dramatically improved (panel D), suggesting that the tissue in which TLR7 agonist was most concentrated was in fact the tissue in which the microscopic morphology was most normal. Taken together, these data suggest that targeting of TLR7 agonist to FRβ⁺ macrophages in fibrotic lungs can prevent fibrosis without systemic activation of the immune system that otherwise limits the use of TLR7 agonists in humans to topical applications (Savage *et al*, 1996; Harrison *et al*, 2004; Geller *et al*, 2010; Biffen *et al*, 2012).

That the targeted TLR7-54 will also have far fewer systemic toxicities than its nontargeted counterpart is shown in Fig 7, where the systemic levels of toxic cytokines and body weight changes are compared. Thus, intravenous administration of nontargeted TLR7-54 to healthy mice induced a dramatic elevation in plasma levels of IL-6, IFNα, and TNFα (Fig 7A–C) which was accompanied by a rapid body weight loss (Fig 7G), whereas FA-TLR7-54 induced little or no systemic release of these cytokines and no body weight loss was detected throughout the study (Fig 7A–C and G). To more thoroughly characterize the concentration range over which FA-TLR7-54 can be administered without induction of systemic inflammation, we next measured the levels of systemic inflammatory cytokines as a function of the dose of FA-TLR7-54. As seen in Fig 7D–F, even a 20 nmol FA-TLR7-54 per mouse stimulated much less inflammatory cytokine release than half the dose of nontargeted TLR7-54. These data suggest that TLR7 agonists can be safely employed to reprogram fibrotic lung macrophages to an anti-fibrotic state if they are targeted to the pulmonary macrophages with a folate receptor targeting ligand.

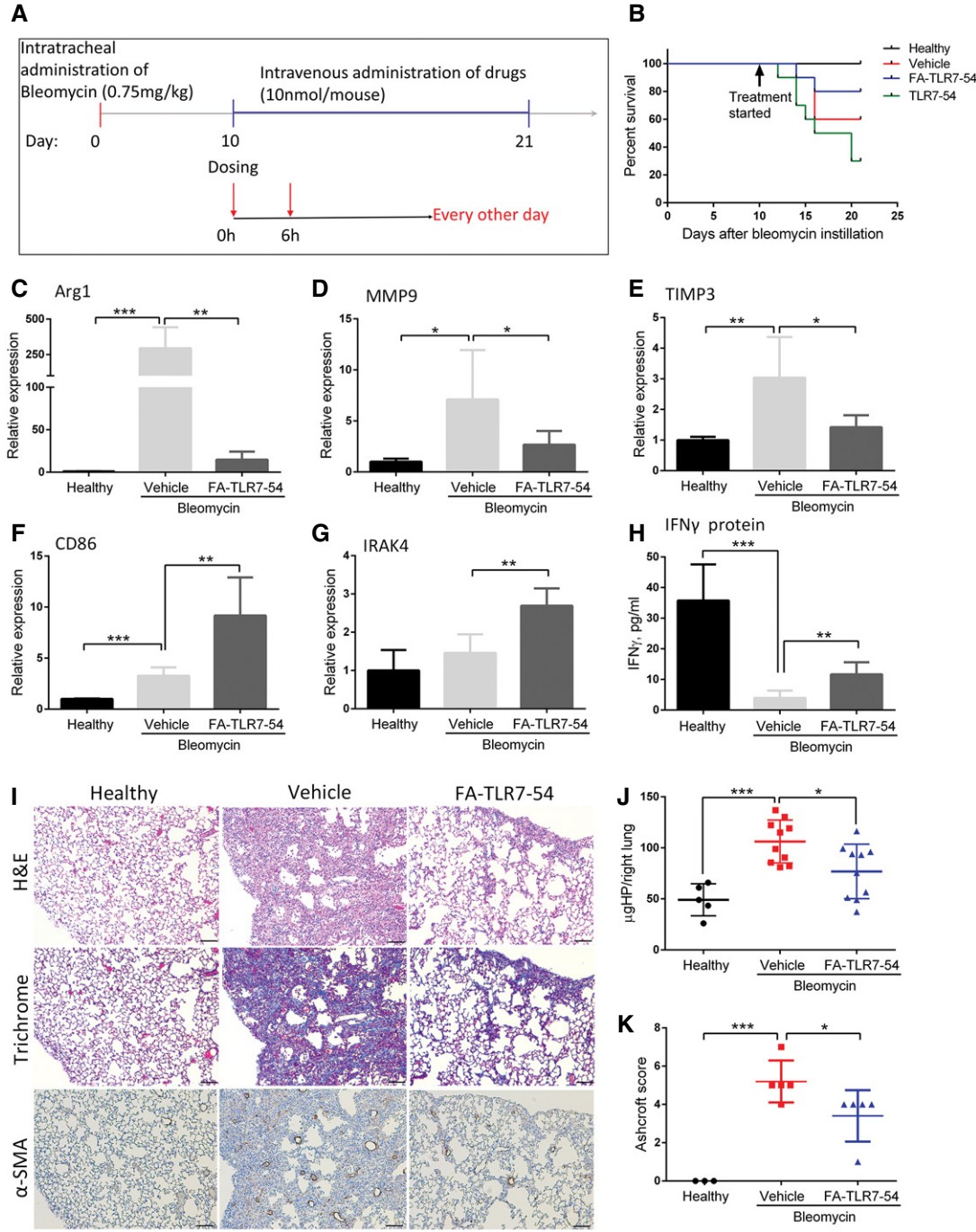

**Figure 5. Effect of alternate-day dosing with FA-TLR7-54 on fibrotic markers in BLM-induced mice.**

A    BLM-induced mice were treated every other day beginning on day 10 with 10 nmol/dose FA-TLR7-54 or TLR7-54.

B    Survival analysis of the treatment groups (healthy, $n = 5$; others, $n = 10$).

C–H    BALF was collected on day 21 and centrifuged to isolate cell pellets. Cells (primarily macrophages) were analyzed by qPCR for Arg1 (C), MMP9 (D), TIMP3 (E), CD86 (F), and IRAK4 (G) ($n = 5$). BALF supernatant was analyzed by ELISA for IFNγ (H) ($n = 5$).

I–K    Lungs were resected and subjected to hematoxylin–eosin (H&E), trichome (collagen), and α-SMA IHC staining (scale bars, 200 μm), (J) or hydrolyzed and analyzed for hydroxyproline content as a measure of collagen content (healthy, $n = 5$; others, $n = 10$), (K) Ashcroft score quantitation of fibrosis ($n = 5$).

Data information: Mean ± SD. Statistical significance between groups was compared using unpaired two-tailed *t*-test (*$P < 0.05$, **$P < 0.01$, ***$P < 0.001$).
Source data are available online for this figure.

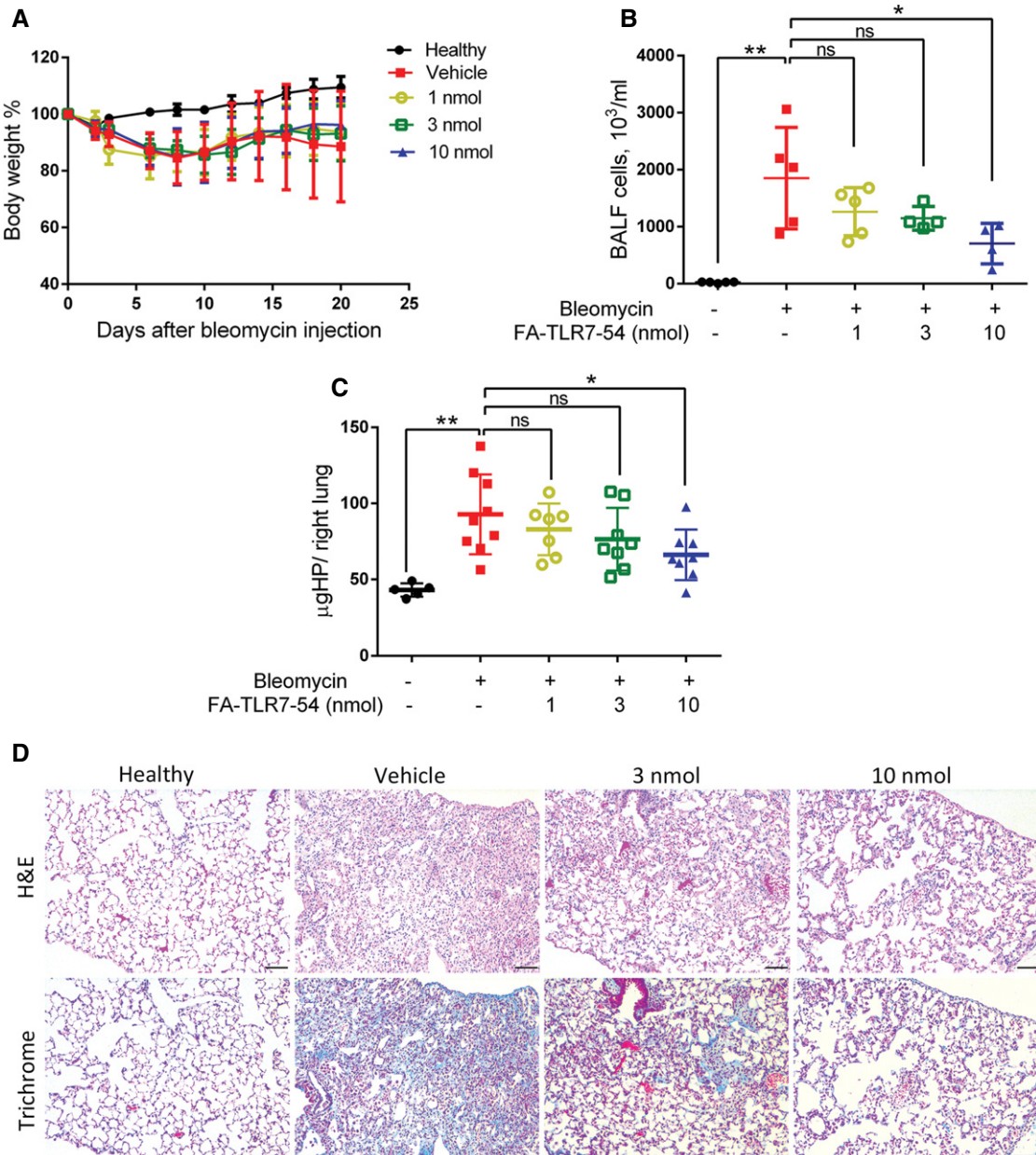

**Figure 6. Concentration dependence of FA-TLR7-54 suppression of BLM-induced fibrosis.**

Healthy control (filled circles) or BLM-induced mice were treated with vehicle (filled squares), 1 nmol (empty circles), 3 nmol (empty squares), or 10 nmol (filled triangles) FA-TLR7-54 and then sacrificed on day 21 for analysis.

A   Analysis of body weight change versus time ($n = 10$).
B   Analysis of the number of cells per milliliter of BALF ($n = 5$).
C   Quantitation of total hydroxyproline content per right lung (healthy control, $n = 5$; others, $n = 7–9$).
D   H&E staining and trichrome staining of lung tissue (scale bars, 200 µm).

Data information: Mean $\pm$ SD. Statistical significance between groups was determined using unpaired two-tailed *t*-test (*$P < 0.05$, **$P < 0.01$).
Source data are available online for this figure.

## Discussion

In this study, we have exploited the fact that FRβ is expressed on profibrotic macrophages but absent from almost all normal tissues to target a TLR7 agonist to fibrotic lung macrophages, thereby avoiding the toxicity that has prevented any clinical use of systemically administered TLR7 agonists (Savage *et al*, 1996; Harrison *et al*, 2004; Geller *et al*, 2010; Biffen *et al*, 2012). Our data demonstrate that delivery of TLR7-54 to these lung macrophages converts the macrophages from an M2-like to an M1-like phenotype. Accompanying this conversion is a decrease in cytokines that stimulate collagen synthesis (i.e., CCL18 and IL-1β) and an increase in

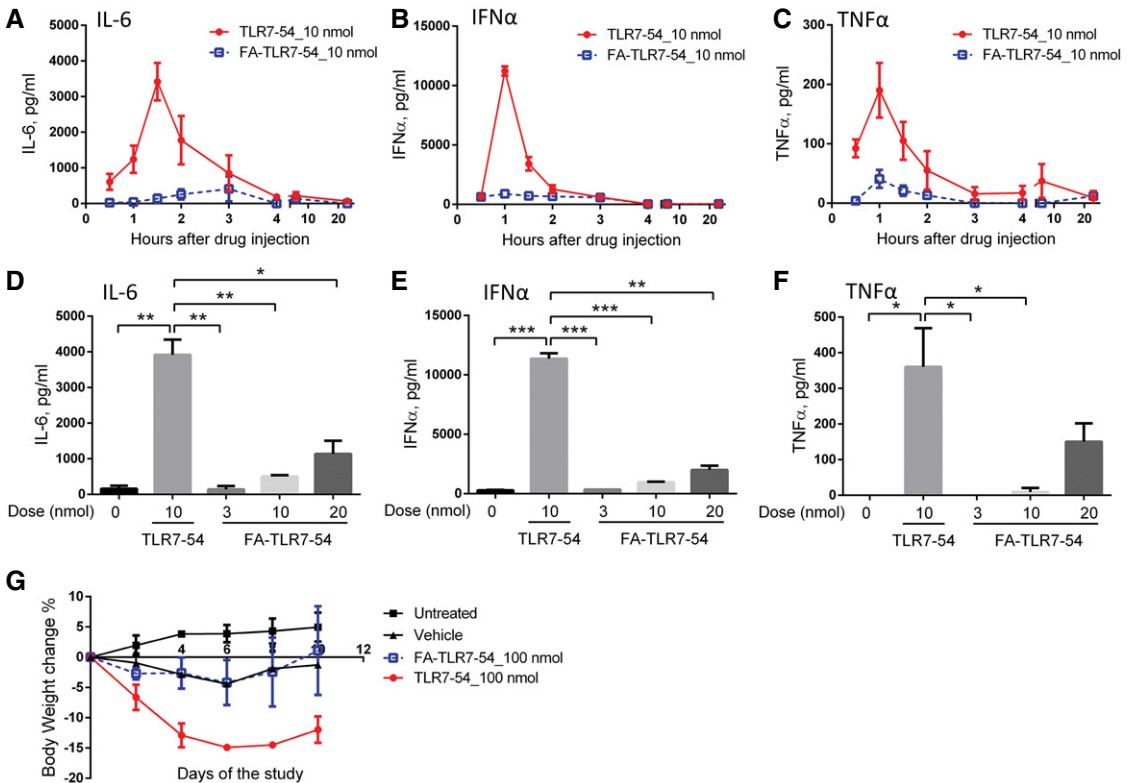

**Figure 7. Comparison of plasma cytokine levels in healthy mice following treatment with nontargeted versus folate-targeted TLR7-54.**

Healthy mice were tail vein injected with 10 nmol TLR7-54 (circles) or FA-TLR7-54 (squares), and peripheral blood was collected at indicated time points after drug injection.

A–C   Measurement of plasma IL-6 (A), IFNα (B), and TNFα (C) ($n$ = 3).

D–F   Effect of drug concentration on plasma levels of IL-6 (D), IFNα (E), and TNFα (F) at 1.5 h, 1 h, or 1 h after treatment, respectively ($n$ = 2).

G   Change in body weight as a measure of systemic toxicity during alternate-day dosing ($n$ = 2).

Data information: Mean ± SD. Statistical significance between groups was compared using unpaired two-tailed $t$-test (*$P$ < 0.05, **$P$ < 0.01, ***$P$ < 0.001).

cytokines that inhibit collagen production (i.e., CXCL10, IFNα, and IFNγ). Noteworthy, unlike the classic effect of TLR7 agonists, the folate-targeted TLR7-54 does not stimulate the secretion of detrimental soluble TNFα (Oikonomou *et al*, 2006). The net consequence of this reprogramming is an increase in alveolar air sacs, decrease in extracellular matrix deposition, and reduction in hydroxyproline/collagen biosynthesis. Because all of these benefits occur without evidence of overt toxicity, we conclude that targeted delivery of TLR7 agonists has the potential to treat fibrotic lung diseases with limited side effects. While more detailed toxicology studies will have to be conducted to confirm these conclusions, the observation that the folate-conjugated fluorescent dye localizes almost exclusively to the fibrotic lungs argues strongly that any TLR7-54 that is not captured by the fibrotic lungs will be minimal.

The entire TLR7 agonist targeting approach was obviously predicated on the observation that FRβ is solely expressed on activated myeloid cells (e.g., predominantly activated monocytes and macrophages) (Feng *et al*, 2011). Thus, no nonmyeloid cells in the lungs, liver, spleen, heart, brain, muscle, intestines, pancreas, bladder, etc., have ever been reported to express FRβ (Ross *et al*, 1994), and even quiescent tissue-resident macrophages that are abundant throughout the body are predominantly FRβ-negative (Xia *et al*, 2009). In fact, all data to date demonstrate that FRβ is only induced

in cells of myelogenous origin following exposure to anti-inflammatory or pro-inflammatory stimuli (Puig-Kröger *et al*, 2009). As a consequence, images obtained in humans with folate-targeted imaging agents commonly display uptake solely in inflamed tissues, malignant lesions, and the kidneys (Low *et al*, 2007; Mahalingam *et al*, 2018). The fact that the FRβ is most prominently expressed on the Mono-AMs and that this macrophage subpopulation contributes disproportionately to the development of lung fibrosis (Misharin *et al*, 2017) supports the contention that delivery of anti-fibrotic drugs using an FRβ targeting strategy should treat fibrosis without damaging healthy tissues. This restricted expression of FRβ to activated myeloid cells is clearly fundamental to our ability to systemically administer a potent TLR7 agonist without systemically activating the immune system.

Unfortunately, the mechanisms of the only two drugs approved by the FDA for the treatment of IPF are not well characterized. While nintedanib is known to inhibit platelet-derived growth factor receptor, fibroblast growth factor receptor, and vascular endothelial growth factor receptor (Richeldi *et al*, 2014), how inhibition of these tyrosine kinases mediates suppression of fibrosis has not been established. Surprisingly, even less is known about the mode of action of pirfenidone, except that it somehow inhibits fibroblast proliferation and production of profibrotic cytokines (e.g., TGFβ

and TNFα) (Noble *et al*, 2011). In the case of FA-TLR7-54, we envision a simple mechanism involving (i) delivery of TLR7-54 to FRβ-expressing profibrotic macrophages via folic acid, (ii) internalization of FA-TLR7-54 into TLR7 containing endosomes by FRβ-mediated endocytosis (Varghese *et al*, 2014), (iii) binding of FA-TLR7-54 to TLR7 in these endosomes (Petes *et al*, 2017), and (iv) activation of pro-inflammatory pathways that reprograms the macrophage from a profibrotic to anti-fibrotic phenotype. Because reprogramming of the profibrotic macrophages inhibits their release of cytokines that stimulate fibroblasts to produce collagen and other extracellular matrix proteins, the consequence is a suppression of fibrosis. In this respect, it is interesting to note that an anti-FRβ-linked pseudomonas exotoxin conjugate has been previously observed to reduce development of fibrosis in BLM-treated mice by depleting their FRβ-expressing macrophages (Nagai *et al*, 2010).

Although the ability of a folate-targeted TLR7 agonist to treat fibrosis was only examined in a murine model of lung fibrosis, we envision that the same strategy might also prove beneficial in treatment of fibrotic diseases of the liver, skin, heart, and the kidneys (Hulsmans *et al*, 2016; Wynn & Vannella, 2016; Tang *et al*, 2019). Most fibrotic pathologies appear to begin with an unknown trauma or insult to an epithelium (Wynn & Vannella, 2016). In response to the resulting tissue damage, released chemokines and other factors promote the infiltration of immune cells (Wynn & Vannella, 2016; Desai *et al*, 2018; Scott *et al*, 2019), including monocytes and macrophages that assume an M2-like phenotype and release profibrotic cytokines (Prasse *et al*, 2006; Wilson *et al*, 2010; Byrne *et al*, 2016; Misharin *et al*, 2017). The chronic secretion of these cytokines then activates tissue-resident or infiltrating fibroblasts/fibrocytes to become myofibroblasts that in turn secrete collagen and other extracellular matrix proteins that stiffen the lung. Given that M2-like macrophages may perform similar fibroblast-activating functions in other traumatized tissues (Hulsmans *et al*, 2016; Wynn & Vannella, 2016; Tang *et al*, 2019), it is not inconceivable that a folate-targeted TLR7 agonist might similarly suppress fibrosis in other tissues also. It will be important to test the impact of FA-TLR7-54 and other folate-targeted anti-fibrotic drugs in other validated models of human fibrotic diseases.

# Materials and Methods

### Differentiation and polarization of M2-like macrophages

Unless otherwise specified, folate-free RPMI 1640 medium containing 10% of heat-inactivated fetal bovine serum (FBS) and 1% penicillin/streptomycin (Invitrogen) was used for all cell culture studies. THP-1 cells (ATCC, TIB-202™) were differentiated into macrophages as described by Genin *et al* (2015) and then polarized to M2-like macrophages as outlined by Fernando *et al* (2014).

Human peripheral blood mononuclear cells (PBMCs) were isolated from fresh peripheral blood and cultured for 2 h in monocyte attachment medium (PromoCell) at a density of 1 million/cm$^2$, after which the cells were washed 4× with pre-warmed PBS and differentiated for 7 days into unpolarized macrophages in RPMI medium containing 20 ng/ml of recombinant human macrophage

colony-stimulating factor (M-CSF) (BioLegend). Polarization of the resulting macrophages into M2-like macrophages was conducted for 48 h as described above.

### Analysis of TLR7 agonist reprogramming of M2-like into M1-like macrophages

Polarized M2-like macrophages described above were treated with either TLR7-54 or FA-TLR7-54, after which the culture medium was analyzed for markers of M1 polarization using ELISA and the cultured cells were evaluated by qPCR for changes in gene expression.

### Mice

Eight-week-old C57BL/6 male mice from Charles River (average weight 22–25 g) were housed under pathogen-free conditions at room temperature (22°C) using a 12 h light–dark cycle. Mice were placed on a folate-deficient chow (Teklad Envigo) upon arrival and acclimated for 1 week prior to initiation of experimental procedures. Freshwater and folate-deficient diet were freely available. All animal procedures were approved by the Purdue Animal Care and Use Committee (PACUC) in accordance with NIH guidelines.

### BLM-induced pulmonary fibrosis

Mice were anesthetized with ketamine/xylazine, their necks were shaved and sterilized, and a small incision was made to expose the trachea. Mice were injected intratracheally with 100 µl sterile PBS or BLM (Cayman Chemicals) dissolved in PBS (0.75 mg/kg). Body weights were monitored every other day. Mice were randomized according to their body weight before starting therapy.

### Characterization of FRβ expression on different pulmonary macrophage subtypes

Ten days after instillation of BLM or PBS, mice were sacrificed and lungs were cannulated, excised, digested to obtain single cells as described below, and then analyzed by flow cytometry to determine FRβ expression on different macrophage subtypes. For *in vivo* folate staining, 10 days post-BLM instillation, mice were tail vein injected with 10 nmol (for *in vivo* imaging) or 100 nmol (for flow cytometric analysis) of OTL38 with or without 200-fold excess of FA-glucosamine. Two hours later, mice were sacrificed and organs were resected and imaged using an AMI live imager (Spectral Imaging). For the identification of the cells that take up OTL38, lungs were harvested immediately following euthanasia, and digested and analyzed by flow cytometry for OTL38-containing cells as described below.

### Treatment of mice with FA-TLR7-54

For *in vivo* reprogramming studies, 10 days post-BLM instillation, mice were tail vein injected with a single dose (10 nmol) of either FA-TLR7-54 or TLR7-54. One or 4 h later, mice were sacrificed and BALF was collected, and lungs were digested for cell sorting, as described below. For therapy studies, drug was intravenously injected every other day beginning on day 10 in the morning and again 6 h later. BALF and lungs were harvested on day 21 and evaluated as described below.

## Analysis of systemic cytokine release and toxicity

Healthy C57BL/6 male mice were tail vein injected with a single dose of either FA-TLR7-54 or TLR7-54, and peripheral blood was collected 0.5, 1, 1.5, 2, 3, 4, 6, and 23 h after the drug injection. Blood was centrifuged at 1,000 $g$ for 10 min, and plasma was collected for ELISA. For the toxicity analyses, healthy C57BL/6 male mice were tail vein injected with 100 nmol of either FA-TLR7-54 or TLR7-54 every other day as described above, and body weight was simultaneously monitored.

## BALF collection and lung digestion

BALF was collected (Sun et al, 2017) and centrifuged at 350 $g$ for 5 min at 4°C, and the supernatant was stored at −80°C for cytokine/chemokine analyses. Cell pellets were depleted of erythrocytes by ammonium chloride lysis, allowed to adhere to culture plates, washed to remove nonadherent cells, and used for qPCR analysis. Final purity was 81%.

Right lungs were resected for subsequent digestion (for flow cytometric analyses) or hydrolysis (for hydroxyproline content analyses), and left lung was fixed in 10% formalin for histological analyses. Right lungs were digested with Lung Dissociation Kit using a gentleMACS Dissociator (Miltenyi Biotec) and then filtered through a 70-μm cell strainer. Cells were collected and depleted of erythrocytes, and the resulting single-cell suspensions were used for flow cytometric analysis.

## Flow cytometric analysis

Cells were treated for 15 min with human or mouse TruStain FcXucodep™ (1:100; BioLegend, #422301 and #101320, respectively) to block Fc receptors and then stained for 30 min on ice with the desired fluorophore-conjugated antibodies purchased from BioLegend (unless otherwise specified): Zombie live/dead viability dye (1:200; #423101), 7-AAD Viability Dye (1:70; #420403), anti-mouse FITC-CD11b (1:100; #101205), anti-mouse PE-F4/80 (1:100; #123109), anti-mouse BB515-CD45 (1:800; Fisher Scientific, #BDB564590), anti-mouse PE-CD64 (1:100; #139304), anti-mouse PerCp/Cy5.5-CD11b (1:400; #101228), anti-mouse PE/Cy7-Ly6C (1:200; #128018), anti-mouse BV605-Ly6G (1:100; #127639), anti-mouse BV421-Siglec F (1:200; Fisher Scientific, #BDB562681), anti-mouse APC-FRβ (1:150; #153306), and anti-human FITC-m909 (10 μg/ml) (Feng et al, 2011). Finally, samples were washed twice, resuspended in FACS buffer, and examined using BD LSRFortessa cell analyzer or Attune NxT flow cytometry, and data were analyzed by BD Accuri C6 Software or FlowJoucodep™ v10. Cell sorting was performed on a BD FACSAria III Cell Sorter, and the sorted macrophages were analyzed for marker genes by qPCR. Macrophage purity was > 95%.

## qPCR analysis of RNA

Macrophage RNA was isolated using Quick-RNAucodep™ Micro-Perp Kit (Zymo Research), and RNA samples were reverse-transcribed using reverse transcription kits (Applied Biosystems). qPCR was performed using the iTaqTM Universal SYBR Green SuperMix (Bio-Rad Laboratories), iCycler thermocycler, and iCycler

**The paper explained**

**Problem**

Medium survival of idiopathic pulmonary fibrosis (IPF) patients following diagnosis is only 2–3 years. While current therapies may mitigate symptoms or slow advancement of the disease, no current therapies can reverse existing fibrosis nor halt disease progression. Given the high probability of mortality, there is an urgent need for the development of improved therapies. Although suppression of profibrotic macrophages can be achieved in vitro by treatment with toll-like receptor 7 (TLR7) agonists, TLR7-based therapies have proven to be too toxic to administer systemically.

**Results**

Unique expression of folate receptor β (FRβ) on fibrotic lung macrophages has enabled the use of folate to target an attached TLR7 agonist specifically to fibrotic lung macrophages, leading to protection of mice against bleomycin-induced lung fibrosis. This remarkable therapeutic benefit is shown to be achieved by reprogramming of the profibrotic pulmonary macrophages to anti-fibrotic macrophages. While a nontargeted TLR7 agonist is shown to cause excessive systemic toxicity as evidenced by rapid body weight loss, dramatic cytokine release, and premature mortality, the folate-targeted TLR7 agonist is demonstrated to cause no detectable toxicity.

**Impact**

The results of this work demonstrate that selective reprogramming of profibrotic macrophages by a folate-targeted TLR7 agonist can alleviate the symptoms of bleomycin-induced pulmonary fibrosis without inducing detectable toxicity. FA-TLR7-54 therefore constitutes an anti-fibrotic agent worthy of further examination for the nontoxic treatment of pulmonary fibrosis.

iQ 3.0 software to track the expression of macrophage polarization markers. Primer sequences for qPCR are shown in Appendix Table S1. Melting curve analysis was performed to confirm specificity, and nonspecific products were not observed in any of the reactions. Each sample was analyzed in triplicate for each marker.

## Collagen determination using hydroxyproline assay

Lobes of the right lung were placed in a pressure-tight vial (Supelco Inc.) and hydrolyzed with 6 N HCl (10 ml/g, v/w) at 120°C for 3.5 h. The hydrolyzate was cooled and centrifuged at 12,000 $g$ for 15 min, and the supernatant was quantified for hydroxyproline as reported previously (Samuel, 2009).

## Analysis of cytokines and chemokines in cell culture supernatants, BALF and plasma

CCL18, IL-1β, IL-6, and CXCL10 were quantified in macrophage supernatants using a human DuoSet ELISA Development System (R&D Systems), an IL-1β Human ELISA Kit (Thermo Fisher Scientific), an ELISA MAXucodep™ Human IL-6 (BioLegend), and ELISA MAXucodep™ Human CXCL10 (BioLegend) as described by manufacturers. Similar mouse-specific kits were employed for quantitation of mouse cytokines. Nitrite secretion was quantitated using a Griess reagent kit (Thermo Fisher Scientific).

**Histologic analysis of lung sections**

Fixed lungs (see above) were embedded in paraffin, sectioned, and stained with H&E, Masson's trichrome or F3 (anti-mouse FRβ antibody, 10 μg/ml; ref. Hu *et al*, 2019), α-SMA (1:100; Abcam, #ab5694), F4/80 (1:100; Bio-Rad, #MCA497), and CD206 (1:2,000; Abcam, #ab64693). Images were obtained using a Leica Versa 8 whole-slide scanner. Tissue sections were examined in a blinded manner by a licensed pathologist. More than $90 \times 10^6$ cells were quantified per section using Aperio Image Scope (Leica Biosystems).

**Statistical analysis**

Statistical analyses were performed with GraphPad Prism 6.0 and Excel. Differences between two unpaired groups were analyzed using the unpaired two-tailed *t*-test. Differences between multiple groups and a control group were analyzed using Dunnett's multiple comparison test. Differences between multiple paired groups were analyzed using Tukey's multiple comparison test as indicated in the figure legends. The exact *P*-values are listed in Appendix Table S2. A *P*-value of < 0.05 was considered significant.

# Data availability

Additional data are provided in the Appendix and are available online.

**Expanded View** for this article is available online.

## Acknowledgements
This study was supported by a gift from Three Lakes Partners, LLC. We thank Victor Bernal-Crespo and the Purdue University Histology Research Laboratory for staining lung samples.

## Author contributions
PSL conceived of the studies. FZ, EAA, BW, EP-C, Y-HL, SUH, CN, MS, IOR, and PSL designed the experiments. BW developed and synthesized the TLR7 agonist and its folate conjugate. FZ, Y-HL, EAA, and SR performed the *in vitro* experiments. QL developed the transduced THP-1 cell line and performed the related *in vitro* experiments. KT trained FZ to develop the murine pulmonary fibrosis model. FZ, EP-C, SUH, SDL, and SR performed the animal experiments. EAA characterized the FRβ expression of macrophage subtypes in murine BLM model. AC performed the pathology evaluation. FZ and PSL analyzed the data. IOR, CN-N, EAA, MS, and PSL provided critical advice. FZ, MS, and PSL wrote the manuscript draft, which was then further refined by IOR and PSL.

## Conflict of interest
The authors declare that they have no conflict of interest.

## For more information
(i) Dr. Philip S. Low's website: https://www.chem.purdue.edu/low/research/index.html

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
