## [Review Process File · EMBO Molecular Medicine]

Reprogramming of Profibrotic Macrophages for Treatment of Bleomycin-Induced Pulmonary Fibrosis

Fenghua Zhang, Ehab Ayaub, Bingbing Wang, Estela Puchulu-Campanella, Yen-Hsing Li, Suraj Hettiarachchi, Spencer Lindeman, Qian Luo, Sasmita Rout, Madduri Srinivasarao, Abigail Cox, Konstantin Tsoyi, Cheryl Nickerson-Nutter, Ivan Rosas, and Philip Low

DOI: [10.15252/emmm.202012034](https://doi.org/10.15252/emmm.202012034)

Corresponding author(s): Philip Low (plow@purdue.edu)

Review Timeline:

Submission Date:	15th Jan 20
Editorial Decision:	2nd Mar 20
Revision Received:	5th May 20
Editorial Decision:	26th May 20
Revision Received:	30th May 20
Accepted:	3rd Jun 20

Editor: Zeljko Durdevic

Transaction Report:

2nd Mar 2020

Dear Dr. Low,

Thank you for the submission of your manuscript to EMBO Molecular Medicine. We have now heard back from the referees whom we asked to evaluate your manuscript. As you will see from the reports below, the referees acknowledge the interest of the study. However, they raise some concerns that should be addressed in a major revision of the present manuscript.

We would welcome the submission of a revised version within three months for further consideration and would like to encourage you to address all the criticisms raised to improve conclusiveness and clarity. Acceptance of the manuscript will entail a second round of review. Please note that EMBO Molecular Medicine encourages a single round of revision only and therefore, acceptance or rejection of the manuscript will depend on the completeness of your responses included in the next, final version of the manuscript. For this reason, and to save you from any frustrations in the end, I would strongly advise against returning an incomplete revision.

I look forward to receiving your revised manuscript.

Yours sincerely,

Zeljko Durdevic

Zeljko Durdevic
Editor

***** Reviewer's comments *****

Referee #1 (Remarks for Author):

The study presented by Zhang et. al. aims to develop a new method of specifically reprogramming M2-like profibrotic macrophages by targeting them with a TLR7 agonist and administering this drug through the folate receptor specifically into activated myeloid cells resulting in a shift to fibrosis-suppressing macrophages. Data shown by authors confirms the specificity of the delivery method through use of FR, a method of targeted therapy widely used in cancer and inflammation. However, authors seem to pay little attention to important cell subtypes implicated in the development of fibrosis such as fibroblasts and alveolar epithelial cells. Additionally, characterization of macrophage polarization was not thoroughly done opting for the use of a handful of polarization markers.

Major concerns:

1. Macrophage polarization can include a spectrum of various subsets, classification of macrophage

- M1 and M2 phenotype by authors using only CD206 seems an oversimplification. To better characterize human M2 macrophage polarization CD163, IL10 among others should be added.
2. In Fig. 2 what does the percentage of CCL18 and IL1- β means? If cytokine production was measured with ELISAs why are results not shown as pg/ml as previously done in other parts of the manuscript?
 3. Number of experimental replicates (in the case of THP1 cells) or biological replicates (in the case of primary macrophages for in vitro results are not clearly stated. Data should represent 3 independent experiments.
 4. Regarding macrophages isolated from BALF: How "pure" was this cell fraction? Which antibody was used to sort these cells? CD68 should be used to sort these cells.
 5. In Fig. 5 macrophages isolated from mice appeared to have reduced CD206 expression however it is not significant. Are other M2 markers more significantly reduced in order to definitively confirm the reprogramming of these cells? Was there any FACS analysis done for these cells?
 6. Authors focused on macrophage polarization but failed to characterize the state of other relevant cells in fibrosis, such as fibroblasts or alveolar epithelial cells after successful FA-TLR-54 treatment. Are there any changes in markers such as α -SMA after macrophage reprogramming?
 7. While in a bleomycin mouse model fibrosis is a separate and exclusive event, in the case of human patients there exists an important connection between lung cancer and fibrosis. Patient cases would challenge specific usage of such folate receptors in fibrosis due to the well-known effect on tumor cells. How would such therapies occur in the case of cancer-associated -fibrosis?
 8. Assessing fibrosis in vivo requires not only quantitation of total hydroxyproline content but also other functional measurements such lung compliance, tissue resistance or fibrotic score.
 9. Immunostaining of M1 and M2 markers in mouse tissue could be done to further asses the reprogramming of macrophages after FA-TLR-54 treatment.
 10. In Fig. 8 what are the basal cytokine levels in plasma of healthy or vehicle mice? Why have controls been omitted from this figure?
 11. Primer sequence listed for IRAK-M actually corresponds to gene NM_029926.5 IRAK4, not IRAK-M (IRAK3), this mistake should be corrected.
 12. There are some typos in the manuscript such as, Bod5y or double parenthesis. References are written in a different font from the rest of the manuscript, there should be consistency throughout the entire text.

Referee #2 (Remarks for Author):

The manuscript by Zhang et al. entitled "Targeted reprogramming of profibrotic macrophages for treatment of IPF" uses expression of folate receptor beta, which is highly expressed on IPF macrophages, but not normal macrophages to deliver a TLR7 agonist in a cell specific manner in an effort to reprogram M2 macrophages to a more M1 like and anti-fibrotic phenotype. The manuscripts compares non-targeted TLR7 agonists to folate-targeted TLR7 agonist in both human cell lines and the murine bleomycin model. The advance in this folate-TLR7 agonist treatment is that it reduces the systemic toxicity of non-targeted TLR7 agonists in vivo. In general the studies are well done showing dose effects that limit the accumulation of fibrosis in the lung and blocking of the effect with excess folate. Importantly, there is a therapeutic effect showing benefit when giving the drug starting on day 10 post-bleomycin.

Major Comments

1) Much of the recent literature in lung fibrosis research has focused on the unique macrophage subsets that are recruited to the lung. The flow panels in Fig. 4 do not allow for discrimination of alveolar macrophages from interstitial macrophages from monocyte-derived alveolar macrophages.

Is the folate receptor b upregulated on all of these different subsets. It would be interesting to know if particular, or all lung macrophages are capable of taking up the targeted TLR7 agonist and if the drug is equally effective in each of these subtypes.

2) The effects of the agonist are inferred to be mediated by TLR7 activation but this is not strictly proven. Does the FA-TLR7-54 compound show efficacy in TLR7-deficient mice?

3) The 10 nM dose of FA-TLR7-54 is most effective, but the histology in Fig. 7D suggest emphysema-like changes and airspace enlargement. This should be quantified to know if this is a limiting toxicity.

Minor Comments

1) I don't think rigidification and rigidify are accepted words in English. Please consider stiffening or stiffen or fibrotic as alternatives.

2) The title is a bit overstated. This manuscript test treatment of bleomycin-induced lung fibrosis, not IPF. Please consider a revision.

Referee #3 (Remarks for Author):

The authors have designed and tested an approach to targeting macrophages in vivo which takes advantage of folate receptor beta (FR beta) expression on activated macrophages. The biologic model employed here is pulmonary fibrosis but targeting macrophages in this fashion is of general interest for a very broad range of medical conditions. This paper therefore has the potential for very wide applicability. Even if this approach is ultimately not fully effective as monotherapy, it might be used as part of a multimodality approach to these diseases. For that reason, this paper should be accepted. However, some additional data and some re-writing should be required.

The strongest part of the paper is the biologic model employed. The authors have shown efficacy and reduced toxicity of their compound relative to a similar but non-targeted compound. The mechanistic component of the paper is less convincing and needs additional data to support the authors' hypotheses as described below. The authors also consistently state that the mechanism of action is via "reprogramming macrophage differentiation from M2 to M1". The authors use a very limited number of markers to make this very broad claim. Furthermore, they use somewhat differing markers for each of their experiments. The authors should consider the following review article when assessing M1 and M2 phenotype in their system: Macrophage Polarization Peter J. Murray Annual Review of Physiology 2017 79:1, 541-566. The authors should also be aware that the classification of macrophages into the M1/M2 dichotomy may not be the best way to consider macrophage heterogeneity specifically with respect to murine bleomycin induced fibrosis (e.g. Aran, D., Looney, A.P., Liu, L. et al. Reference-based analysis of lung single-cell sequencing reveals a transitional profibrotic macrophage. Nat Immunol 20, 163-172 (2019)). A statement of "reprogramming" would imply that some permanence was attached to the effect, which the authors have not shown. Altered differentiation or expression pattern would seem more accurate.

There are also minor stylistic and presentation issues that should be addressed. For example, the paper would be strengthened if the authors would reframe their paper into a more conventional hypothesis driven format. This would help clarify what is actually being shown by the data in each section.

Detailed comments:

The authors show that a TLR7 agonist with and without a folate label have similar have similar

effect on a M2 polarized THP-1 cell line, murine bone marrow derived macrophages and human peripheral blood monocytes in reducing M2 markers and increasing M1 markers (using different markers in each case). For the human cells, folate excess blocked the effect. There is a dose response issue however, in that folate excess should only block folate receptor dependent binding and not direct binding ie there should be a shift in the dose response curve but not complete blockage. Perhaps the FA-TLR7-54 compound has no access to the endosome in absence of folate receptor binding due to the added sidechain? Why is the dose response curve for the THP-1 system unaffected by the presence of the folate on the TLR7 agonist? Is the Kd the same for FR beta as TLR7? Finally, the authors present no direct evidence indicating that their compound requires binding and internalization via FR-beta vs some other folate dependent binding site.

Figure 1B and 2 should be one panel. Figure 1A should be in the supplementary material along with the description of the synthetic pathway and the LC-MS of the compound.

In their bleomycin model, compound uptake is shown to be limited to the lungs which is blocked in presence of folate excess. A key missing data point is extent of co-expression of compound uptake with FR-beta expression. The flow cytometry analysis of uptake in fig 4B is very hard to interpret due to the way it is displayed. More conventional gate analysis with overlay of histograms rather than display of multiple SSC vs stain plots would be helpful. Pharmacokinetics of their compound vs the parent TLR7 agonist in this model should also be included to see if clearance is affected by the folate sidechain.

The authors then analyze bulk populations of lung macrophages, BAL cells and BAL protein levels for relevant mediators including ARG1, CD206, CXCL10, IL-6, IFN alpha and TNF alpha in response TLR7-54 or FA-TLR7-54 (fig 5) at a single time point shortly after administration. Very noteworthy is markedly reduced levels of expression of TNF alpha in the latter group which is only commented on in passing by the authors. One hypothesis is that this is due to stimulation FR beta negative cells by TLR7-54 but this is not further investigated.

The authors then show that repeated administration of FA-TLR7-54 prevents fibrosis when administered on day 10 of their protocol. It is a significant strength of the manuscript that they can show an effect this late after bleomycin administration. There is a dramatic effect on various mediators relevant to fibrosis with a corresponding reduction in total fibrosis at day 21. The authors note that fibrosis resolves in their model as it does in most bleomycin models. The question then is whether their intervention reduces development of fibrosis or accelerates the resolution of fibrosis. The authors imply the latter but have no data on that issue. A time course of fibrosis and histology over time +/- intervention might shed light on that issue.

The authors use the phrase "suppressing the symptoms of fibrosis" when they are actually measuring fibrosis itself. IRAK-M is a marker of TLR signaling suppression, not activation. Histology does not evaluate "tissue density" (not sure what they mean here). Routine histology allows evaluation of features of tissue repair and inflammatory infiltrate.

The dose response curve in figure 7 is of unclear significance with respect to toxicology and adds little to the paper. Figure 8 on the other hand is very helpful to document lack of systemic effect of this compound at doses that are pharmacologically relevant. A dose response curve for systemic expression of these mediators would be important to help establish a therapeutic window for the compound.

***** Reviewer's comments *****

Referee #1 (Remarks for Author):

The study presented by Zhang et. al. aims to develop a new method of specifically reprogramming M2-like profibrotic macrophages by targeting them with a TLR7 agonist and administering this drug through the folate receptor β specifically into activated myeloid cells resulting in a shift to fibrosis-suppressing macrophages. Data shown by authors confirms the specificity of the delivery method through use of FR β , a method of targeted therapy widely used in cancer and inflammation. However, authors seem to pay little attention to important cell subtypes implicated in the development of fibrosis such as fibroblasts and alveolar epithelial cells. Additionally, characterization of macrophage polarization was not thoroughly done opting for the use of a handful of polarization markers.

Major concerns:

1. Macrophage polarization can include a spectrum of various subsets, classification of macrophage M1 and M2 phenotype by authors using only CD206 seems an oversimplification. To better characterize human M2 macrophage polarization CD163, IL10 among others should be added.

In Fig. 2 we now examine 3 additional M2 markers (Arg1, CD206 and CD163). The new data confirms that both TLR7-54 and FA-TLR7-54 change the phenotype from an M2-like to M1-like phenotype. In Fig. 4 we have now added data further demonstrating the effect of FA-TLR7-54 on the reprogramming of lung macrophages in bleomycin-induced mice. Thus, we now characterized the levels of M2-like macrophages by quantitating the changes in three M2 marker transcripts (Arg1, CD206 and CD163) in both the affected lungs and also in the associated bronchoalveolar lavage fluid (BALF). Importantly, treatment with FA-TLR7-54 significantly decreases the levels of all three markers in both fibrotic lung tissues (panel A) and BALF (panel B). These data all support the contention that FA-TLR7-54 reprograms pulmonary fibrotic lung macrophages from an M2-like to M1-like phenotype. Nevertheless, to still further establish this conclusion, we have also characterized the changes in mRNA levels for three well-accepted M1 markers, namely CXCL10, IL-6 and TNF α . In all three cases, treatment with FA-TLR7-54 causes an increase in M1 marker mRNA expression, both in the affected lungs (panel A) and associated BALF (panel B). Then in panel C, we show the changes in some representative protein markers for M1 macrophages in BALF, which also confirm that a change from M2-like to M1-like characteristics is induced by FA-TLR7-54. Taken together, these studies provide very compelling evidence that a repolarization of fibrotic lung macrophages has been induced by FA-TLR7-54. We did not measure the changes in IL-10 levels, since its regulation during fibrosis is very controversial in the literature (Millar, 2006). We have not only added the extra panels in Fig. 2 (A-C) and Fig 4. but have also made the requisite changes to the figure legends and have added brief explanatory text on pages 7 and 10 to describe the new data.

2. In Fig. 2 what does the percentage of CCL18 and IL1- β means? If cytokine production was measured with ELISAs why are results not shown as pg/ml as previously done in other parts of the manuscript?

We have reorganized the former Fig. 2 to Figs. 1 B-C. We thank the reviewer for this observation and have changed the units for the cytokines to pg/ml (now in Figs. 1 B-C).

3. Number of experimental replicates (in the case of THP1 cells) or biological replicates (in the case of primary macrophages for in vitro results are not clearly stated. Data should represent 3 independent experiments.

We apologize for this oversight. We have now indicated the number of independent experiments used to calculate the data points for each experiment in each figure legend (page 30). In no case were fewer than 3 independent experiments used to construct the figures.

4. Regarding macrophages isolated from BALF: How "pure" was this cell fraction? Which antibody was used to sort these cells? CD68 should be used to sort these cells.

In order to obtain a live macrophage fraction from digested fibrotic lungs, we used both Zombie violet and F4/80 for FACS cell sorting and obtained a cell preparation that was >95% macrophages. To enrich BALF cells for macrophages, we centrifuged the fluid to obtain a cell pellet, and after removing the erythrocytes with lysis buffer, we seeded the cells into RPMI 1640 media and allowed the macrophages to adhere for 2 hours. The unattached cells together with the media were removed and the adherent macrophages were detached with Accutase and used for RNA analysis or flow cytometry. Based on the flow cytometry data, the purity was ~81%. We have added these new data to the Methods Section on page 19 (highlighted in yellow).

5. In Fig. 5 macrophages isolated from mice appeared to have reduced CD206 expression however it is not significant. Are other M2 markers more significantly reduced in order to definitively confirm the reprogramming of these cells? Was there any FACS analysis done for these cells?

We repeated the CD206 RNA analysis for the digested fibrotic lung cells in triplicate and obtained data that still show significant differences among the samples. The former Fig.5 was reorganized to Fig.4 and these new data are now included in Fig. 4A. As suggested above, we also quantitated the level of CD163 mRNA in all samples and found that the CD163 was also reduced significantly upon treatment with TLR7-54 or FA-TLR7-54. Taken together and as noted above, it can be concluded that FA-TLR7-54 downregulates M2 markers (Arg1, CD206 and CD163) and upregulates M1 markers (CXCL10, IL-6 and TNF α) in vivo, demonstrating that it indeed repolarizes fibrotic lung macrophages in vivo. We did not examine the cell surface markers by flow cytometry because each of the different M1 and M2 markers rise and fall in expression levels following FA-TLR7-54 treatment over different time scales, whereas mRNA levels all seem to change with similar rapid kinetics.

6. Authors focused on macrophage polarization but failed to characterize the state of other relevant cells in fibrosis, such as fibroblasts or alveolar epithelial cells after successful FA-TLR7-54 treatment. Are there any changes in markers such as α -SMA after macrophage reprogramming?

We have now examined the changes in alpha-smooth muscle actin (α -SMA) in fibrotic lung fibroblasts by IHC (see Fig. 5I), and these data demonstrate a significant reduction in α -SMA following treatment with FA-TLR7-54. Because α -SMA in fibrotic lungs is only found in activated (not quiescent) fibroblasts and vascular smooth muscle cells (which are easily distinguished), the data shown in the Fig.5I demonstrate that FA-TLR7-54 also causes a decrease in activated fibroblasts. This important result should logically be anticipated (since activated fibroblasts are induced from nonactivated fibroblasts in large part by cytokines released from activated M2-like macrophages), but it is nice to firmly demonstrate it in fibrotic lung tissue from live mice. We have added a sentence to page 12 of the Results Section to describe the new data.

7. While in a bleomycin mouse model fibrosis is a separate and exclusive event, in the case of human patients there exists an important connection between lung cancer and fibrosis. Patient cases would challenge specific usage of such folate receptors in fibrosis due to the well-known effect on tumor cells. How would such therapies occur in the case of cancer-associated -fibrosis?

This is an insightful question. As recognized by the reviewer, accessible folate receptors are expressed in significant numbers on three different cells types, namely cancer cells, activated macrophages (including tumor-associated macrophages) and the proximal tubule cells of the kidneys. Because the proximal tubule cells of the kidneys do not express TLR7, uptake of FA-TLR7-54 should have no effect on cell properties. In the case of tumor-associated macrophages, we have found that they are also reprogrammed from an M2-like to M1-like phenotype (i.e. similar to fibrotic lung macrophages) with the treatment of FA-TLR7 agonist, rendering the tumor microenvironment much more sensitive to immunotherapies (including CAR T cell therapies). This change can be very beneficial, since it can be exploited to improve many cancer therapies. In contrast, the effect of FA-TLR7-54 on the folate receptor expressing cancer cell may be harmful, since data in the literature demonstrate that a TLR7 agonist can promote cancer cell survival in those cancers that express TLR7 (Cherfils-Vicini et al, 2010). Based on these data, we would conclude that before administering a FA-TLR7 agonist to pulmonary fibrosis patients, one should assure that the patient does not have a folate receptor positive/TLR7 positive cancer.

Although we would like to add a paragraph to the Discussion Section summarizing these considerations, we have refrained from doing so because we are already at the word limit for EMBO Molecular Medicine. However, if the editorial staff disagrees with this decision, we would be happy to add a paragraph summarizing the above considerations.

8. Assessing fibrosis in vivo requires not only quantitation of total hydroxyproline content but also other functional measurements such lung compliance, tissue resistance or fibrotic score.

We have now included the Ashcroft score evaluated by a veterinary pathologist blindly to Fig. 5K. The score of FA-TLR7-54 treated group was significantly decreased compared to vehicle treated group, which is consistent with the hydroxyproline and histological data. We have also made a minor wording change on page 12 (Results Section) and page 32 (Figure legend) to describe the new data.

9. Immunostaining of M1 and M2 markers in mouse tissue could be done to further assess the reprogramming of macrophages after FA-TLR-54 treatment.

We have performed the requested CD206 immunofluorescence staining on the fibrotic lung tissues and the images are now included in the Fig. EV4 now. The pictures demonstrate that CD206 positive macrophages are significantly decreased in FA-TLR7-54 treated fibrotic lungs. We have also added a sentence to Results Section on page 11 to describe the images.

10. In Fig. 8 what are the basal cytokine levels in plasma of healthy or vehicle mice? Why have controls been omitted from this figure?

We have reorganized the former Fig. 8 to Fig. 7. We now have included the basal levels of IL6, IFN α and TNF α in the new panels D, E and F, respectively, of Fig. 7. The basal cytokine levels of IL-6, IFN α and TNF α in plasma of healthy and vehicle-treated mice are very low compared to nontargeted TLR7-54 treated mice.

11. Primer sequence listed for IRAK-M actually corresponds to gene NM_029926.5 IRAK4, not IRAK-M (IRAK3), this mistake should be corrected.

We thank the reviewer for reading our manuscript carefully and pointing out this mistake. We have corrected this mistake to IRAK4 in Fig. 5G, the legend to Fig. 5G (page 32), the Results Section (page 11), and the Appendix (Table S1).

12. There are some typos in the manuscript such as, Bod5y or double parenthesis. References are written in a different font from the rest of the manuscript, there should be consistency throughout the entire text.

Thanks again for reading our manuscript carefully and kindly pointing out these typos and format issues. We have carefully checked through the manuscript and corrected them.

Referee #2 (Remarks for Author):

The manuscript by Zhang et al. entitled "Targeted reprogramming of profibrotic macrophages for treatment of IPF" uses expression of folate receptor beta, which is highly expressed on IPF macrophages, but not normal macrophages to deliver a TLR7 agonist in a cell specific manner in an effort to reprogram M2 macrophages to a more M1 like and anti-fibrotic phenotype. The manuscript compares non-targeted TLR7 agonists to folate-targeted TLR7 agonist in both human cell lines and the murine bleomycin model. The advance in this folate-TLR7 agonist treatment is that it reduces the systemic toxicity of non-targeted TLR7 agonists in vivo. In general the studies are well done showing dose effects that limit the accumulation of fibrosis in the lung and blocking of the effect with excess folate. Importantly, there is a therapeutic effect showing benefit when giving the drug starting on day 10 post-bleomycin.

Major Comments

1) Much of the recent literature in lung fibrosis research has focused on the unique macrophage subsets that are recruited to the lung. The flow panels in Fig. 4 do not allow for discrimination of alveolar macrophages from interstitial macrophages from monocyte-derived alveolar macrophages. Is the folate receptor β upregulated on all of these different subsets. It would be interesting to know if particular, or all lung macrophages are capable of taking up the targeted TLR7 agonist and if the drug is equally effective in each of these subtypes.

We thank the reviewer for this insightful question. We were also very interested in this question, and even collected the desired data, but did not include these data due to manuscript length constraints. The data now shown in Fig. EV3 demonstrate that monocyte-derived alveolar macrophages (but not interstitial macrophages or tissue resident alveolar macrophages) increase significantly following bleomycin instillation (Misharin et al, 2017), and that this subpopulation of macrophages contains the vast majority of folate receptor β positive macrophages in the fibrotic lungs. Thus, ~70% of the macrophages in the fibrotic lungs are monocyte-derived alveolar macrophages, and the FR β expression is primarily concentrated in this subpopulation. These data are important, since the monocyte-derived alveolar macrophages are thought to constitute the major subpopulation responsible for inducing the fibrosis (Misharin et al, 2017). We have also added a sentence to Results Section (page 9) and Discussion Section (page 15) to summarize these data.

2) The effects of the agonist are inferred to be mediated by TLR7 activation but this is not strictly proven. Does the FA-TLR7-54 compound show efficacy in TLR7-deficient mice?

We performed this exact study on THP-1-NF- κ B cells that were either transduced with human TLR7 (hTLR7) or not transduced with hTLR7. We found that TLR7-54 stimulated NF- κ B activation in the THP-1-NF- κ B cells that expressed hTLR7 but failed to activate NF- κ B in the THP-1-NF- κ B cells that lacked hTLR7. These data are now included as Fig. EV2. We have also added 3 sentences on page 6 to explain the data and have modified the Appendix materials and methods accordingly.

3) The 10 nmol dose of FA-TLR7-54 is most effective, but the histology in Fig. 7D suggest emphysema-like changes and airspace enlargement. This should be quantified to know if this is a limiting toxicity.

Thank you for this observation. The former Fig. 7D is reorganized to Fig. 6D now. The following response was provided by the pathologist who read all of the pathology slides. “We have thoroughly assessed all histologic slides for possible co-morbidities or toxicities. Over-inflation of the lungs during fixative infusion is common if the volume is not appropriately adjusted to the in vivo state of inflation. The lesions identified are most consistent with postmortem over-inflation, not emphysema. Emphysema commonly affects entire lobules, terminal bronchioles and alveoli. Additionally, if this were emphysema, then fragments of alveolar wall may be apparent histologically. For these reasons, we are confident that the observed differences in alveolar space in Fig. 6D are due to over-inflation of infused fixative.”. In brief, the pathologist at Purdue University saw no overt signs of toxicity due to treatment with FA-TLR7-54.

Minor Comments

1) I don't think rigidification and rigidify are accepted words in English. Please consider stiffening or stiffen or fibrotic as alternatives.

Thank you for your kind suggestion, we have changed the “rigidification” to “stiffening” on page 3 and “rigidify” to “stiffen” on page 4.

2) The title is a bit overstated. This manuscript test treatment of bleomycin-induced lung fibrosis, not IPF. Please consider a revision.

We have retitled the manuscript to read, “Reprogramming of Profibrotic Macrophages for Treatment of Bleomycin-Induced Pulmonary Fibrosis”.

Referee #3 (Remarks for Author):

The authors have designed and tested an approach to targeting macrophages in vivo which takes advantage of folate receptor beta (FR beta) expression on activated macrophages. The biologic model employed here is pulmonary fibrosis but targeting macrophages in this fashion is of general interest for a very broad range of medical conditions. This paper therefore has the potential for very wide applicability. Even if this approach is ultimately not fully effective as monotherapy, it might be used as part of a multimodality approach to these diseases. For that reason, this paper should be accepted. However, some additional data and some re-writing should be required.

We thank this review for these very generous remarks.

The strongest part of the paper is the biologic model employed. The authors have shown efficacy and reduced toxicity of their compound relative to a similar but non-targeted compound. The mechanistic component of the paper is less convincing and needs additional data to support the authors' hypotheses as described below. The authors also consistently state that the mechanism of action is via "reprogramming macrophage differentiation from M2 to M1".

The authors use a very limited number of markers to make this very broad claim. Furthermore, they use somewhat differing markers for each of their experiments. The authors should consider the following review article when assessing M1 and M2 phenotype in their system: Macrophage Polarization Peter J. Murray Annual Review of Physiology 2017 79:1, 541-566.

We also thank the reviewer for these insightful suggestions. Indeed, this reviewer's major concern is very similar to the two main comments of Reviewer #1. As noted in our response to Reviewer #1, we have now examined three additional M2-like macrophage markers (Arg1, CD206 and CD163) as suggested in the paper (Murray, 2017) to demonstrate that both TLR7-54 and FA-TLR7-54 reduce expression of M2-like polarization markers (see revised Fig. 2). Together with the protein analysis data, which show that the FA-TLR7-54 can suppress the secretion of CCL18 while increasing secretion of CXCL10 and IL6, we can now confidently conclude that FA-TLR7-54 reprograms the lung macrophages from a profibrotic M2-like phenotype to an anti-fibrotic M1-like phenotype.

In Fig. 4 we have also examined the effects of FA-TLR7-54 on the same additional M2 markers (Arg1, CD206 and CD163) in both the affected lungs and in the associated bronchoalveolar lavage fluid (BALF) of the bleomycin-induced mice. Importantly, treatment with FA-TLR7-54 significantly decreases the levels of all three markers in both fibrotic lung tissues (panel A) and BALF (panel B). These data further support the contention that FA-TLR7-54 reprograms pulmonary fibrotic lung macrophages from an M2-like to M1-like phenotype. Nevertheless, to even more conclusively establish this conclusion, we have also characterized the changes in mRNA levels for three well-accepted M1 markers, namely CXCL10, IL-6 and TNF α . In all three cases, treatment with FA-TLR7-54 causes an increase in the M1 marker mRNA expression, both in affected lungs (panel A) and associated BALF (panel B). Then in panel C, we show the changes induced in several representative protein markers for M1 macrophages in BALF, which also confirm that a shift from M2-like to M1-like characteristics is induced by FA-TLR7-54.

Taken together, we believe these studies provide very compelling evidence that a repolarization of fibrotic lung macrophages is induced by FA-TLR7-54. We have not only added the extra panels in Fig. 2 (A-C) and Fig. 4, but also as noted in our response to reviewer #1 we have made minor changes to the associated figure legends and the text on pages 7 and 10. We have also added the Murray et al reference to the list of References.

The authors should also be aware that the classification of macrophages into the M1/M2 dichotomy may not be the best way to consider macrophage heterogeneity specifically with

respect to murine bleomycin induced fibrosis (e.g. Aran, D., Looney, A.P., Liu, L. et al. Reference-based analysis of lung single-cell sequencing reveals a transitional profibrotic macrophage. *Nat Immunol* 20, 163-172 (2019)).

We completely agree with this concern and totally concur that classification of macrophages into an M1/M2 dichotomy is a major over-simplification. Indeed, we find in our own studies that the macrophages in fibrotic tissues really constitute a rainbow of states extending continuously from strongly profibrotic to strongly inflammatory. And some pulmonary fibrosis macrophages even simultaneously express both M1 and M2 markers on the same macrophage. However, there is currently an inadequacy in our vocabulary for describing the infinite range of phenotypes in macrophages, and to minimize the complexities associated with describing the effects of FA-TLR7-54 on the plethora of macrophage phenotypes found in the lungs of bleomycin-induced mice, we have reluctantly resorted to the M1/M2 terminology because we believe the readership understands the inaccuracies in this nomenclature and know of no other non-verbose terminology that is better. Having said all of this, we would be happy to restate these terms in any manner recommended by the reviewer.

A statement of "reprogramming" would imply that some permanence was attached to the effect, which the authors have not shown. Altered differentiation or expression pattern would seem more accurate.

We believe that the FA-TLR7-54 induced changes that we report are both expansive and durable, since they lead to significant alterations in lung biochemistry, composition, and morphology that persist to the end of the 21 day studies. Thus, the data in Figs. 5 and 6 reveal substantial changes in Arg1, MMP9, TIMP3, CD86, IRAK4, IFN γ , α -SMA, hydroxyproline, collagen, H&E staining morphology, and Ashcroft score in the lungs of the bleomycin-instilled mice. This spectrum of alterations seems to us like a reprogramming, since the markers analyzed are found in multiple distinct pathways controlling macrophage behavior. However, if the reviewer feels strongly about changing our terminology here, we would be happy to do so.

There are also minor stylistic and presentation issues that should be addressed. For example, the paper would be strengthened if the authors would reframe their paper into a more conventional hypothesis driven format. This would help clarify what is actually being shown by the data in each section.

We have reworded the concluding paragraph in the Introduction Section (page 4) to reframe the purpose of the manuscript to testing the hypothesis that reprogramming of pulmonary macrophages from a profibrotic to anti-fibrotic phenotype can treat the symptoms of pulmonary fibrosis. We then proceed to conduct studies that test this hypothesis.

Detailed comments:

The authors show that a TLR7 agonist with and without a folate label have similar have similar effect on a M2 polarized THP-1 cell line, murine bone marrow derived macrophages and human

peripheral blood monocytes in reducing M2 markers and increasing M1 markers (using different markers in each case).

For the human cells, folate excess blocked the effect. There is a dose response issue however, in that folate excess should only block folate receptor dependent binding and not direct binding ie there should be a shift in the dose response curve but not complete blockage. Perhaps the FA-TLR7-54 compound has no access to the endosome in absence of folate receptor binding due to the added sidechain? Why is the dose response curve for the THP-1 system unaffected by the presence of the folate on the TLR7 agonist? Is the K_d the same for FR beta as TLR7? Finally, the authors present no direct evidence indicating that their compound requires binding and internalization via FR-beta vs some other folate dependent binding site.

The reviewer's suspected answer to his/her own question is correct when he/she speculates "Perhaps the FA-TLR7-54 compound has no access to the endosome in absence of folate receptor binding due to the added sidechain?". We have indeed designed FA-TLR7-54 to be impermeable to cells unless the cell expresses a folate receptor that can internalize the folate-drug conjugate by folate receptor mediated endocytosis. Because TLR7 is exclusively located in intracellular endosomes, no cell lacking an empty cell surface folate receptor can be stimulated by FA-TLR7-54. For this reason, saturation of all cell surface folate receptors with excess folate-glucosamine is shown to block any activation of TLR7 by FA-TLR7-54. Moreover, in the experiments with cultured THP-1 cells, addition of FA-TLR7-54 yields a similar response to addition of TLR7-54, because the FA-TLR7-54 can enter the cells via their folate receptors and the TLR7-54 will enter the same cells by passive diffusion across the cell's membranes (i.e. TLR7-54 is very membrane permeable). However, in live animals FA-TLR7-54 will be concentrated solely in cells expressing a folate receptor, while TLR7-54 will passively enter all cells of the body, thereby causing off-target toxicity to healthy cells.

Regarding the next question, the binding affinities of FA-TLR7-54 for a folate receptor and TLR7 are fortuitously similar (~5 nM). Finally, in response to the last question in this series, we have been studying folate receptors for 30 years now and have never found an alternative route for folate conjugate uptake into cells other than via a folate receptor. While there are many folate binding enzymes inside cells, none of these occur on the cell surface. They also have low affinity for folate, since they much prefer to bind reduced folates such as THF, 5-MeTHF, DHF, N-formylTHF, etc.

Figure 1B and 2 should be one panel. Figure 1A should be in the supplementary material along with the description of the synthetic pathway and the LC-MS of the compound.

We appreciate this suggestion for reorganizing our figures. We have complied with the suggestion and have moved Fig. 2 into Fig. 1 and have moved former Fig. 1A to Supplementary materials. We now have 7 figures instead of 8 figures in the main manuscript.

In their bleomycin model, compound uptake is shown to be limited to the lungs which is blocked

in presence of folate excess. A key missing data point is extent of co-expression of compound uptake with FR-beta expression. The flow cytometry analysis of uptake in Fig 4B is very hard to interpret due to the way it is displayed. More conventional gate analysis with overlay of histograms rather than display of multiple SSC vs stain plots would be helpful.

We have repeated this study and have replotted the flow cytometry data to show the histograms in addition to the scatter plots (now in Fig. 3). The percent of macrophages in the bleomycin-induced lungs that are folate receptor beta positive is 20%. Moreover, few if any non-macrophage cells are seen to bind folate-dye conjugate, confirming that FR β is only expressed on a subset (activated) of macrophages. We have also made minor modifications to the wording on page 9 (Result) and page 31 (Figure legend) to describe these new data.

Pharmacokinetics of their compound vs the parent TLR7 agonist in this model should also be included to see if clearance is affected by the folate sidechain.

Based on PK data obtained on many other small molecule folate conjugates (8 have entered human clinical trials), we can confidently predict that attachment of folate to TLR7 agonist will not significantly alter its PK. Half-life values in the blood for folate conjugates are usually 20 to 30 minutes.

The authors then analyze bulk populations of lung macrophages, BAL cells and BAL protein levels for relevant mediators including ARG1, CD206, CXCL10, IL-6, IFN alpha and TNF alpha in response TLR7-54 or FA-TLR7-54 (Fig 5) at a single time point shortly after administration. Very noteworthy is markedly reduced levels of expression of TNF alpha in the latter group which is only commented on in passing by the authors. One hypothesis is that this is due to stimulation of FR beta negative cells by TLR7-54 but this is not further investigated.

We thank the reviewer for this insightful question. We agree with the reviewer's hypothesis that nontargeted TLR7-54 will stimulate all cells with TLR7 while FA-TLR7-54 will only enter and stimulate FR beta positive cells. Thus, nontargeted TLR7-54 will create systemic induction of TNF α while our FA-targeted conjugate will only stimulate TNF α production by FR-expressing macrophages. This is the reason we believe that FA-TLR7-54 is both more potent and less toxic than free TLR7-54 in vivo.

The authors then show that repeated administration of FA-TLR7-54 prevents fibrosis when administered on day 10 of their protocol. It is a significant strength of the manuscript that they can show an effect this late after bleomycin administration. There is a dramatic effect on various mediators relevant to fibrosis with a corresponding reduction in total fibrosis at day 21. The authors note that fibrosis resolves in their model as it does in most bleomycin models. The question then is whether their intervention reduces development of fibrosis or accelerates the resolution of fibrosis. The authors imply the latter but have no data on that issue. A time course of fibrosis and histology over time +/- intervention might shed light on that issue.

The question of “whether their intervention reduces development of fibrosis or accelerates the resolution of fibrosis” is a question we have frequently pondered and concluded there is no way to answer. Thus, if we block a step in the pathway leading to “development of fibrosis” we should observe no additional increase in fibrosis but instead observe a decrease in fibrosis as the constitutive degradative processes continue to function. Similarly, if we accelerate the pathways causing degradation of the fibrosis, we should still see a decrease in the fibrosis relative to controls. However, the fact that we repolarize the activated macrophages so that they discontinue secreting profibrotic cytokines/chemokines (i.e. CCL18, IL-1 β) that induce lung fibroblasts to secrete collagen etc. suggests that we are inhibiting development of the fibrosis rather than accelerating its resolution. We could add a paragraph summarizing these arguments if the editors wished.

The authors use the phrase "suppressing the symptoms of fibrosis" when they are actually measuring fibrosis itself.

We have changed the wording as suggested.

IRAK-M is a marker of TLR signaling suppression, not activation.

We thank the reviewer for reading our manuscript carefully and pointing out this mistake. The primer sequence in the list was correct, which corresponds to gene NM_029926.5 IRAK4, but we accidentally wrote the name as IRAK-M. We have corrected the name to IRAK4 in the Fig. 5G, Figure legend (page 32), Result Section (page 11), and the Appendix table S1.

Histology does not evaluate "tissue density" (not sure what they mean here). Routine histology allows evaluation of features of tissue repair and inflammatory infiltrate.

We thank the reviewer for his suggestion on more accurate wording. We have changed the “tissue density” to “tissue morphology” on page 11.

The dose response curve in Figure 7 is of unclear significance with respect to toxicology and adds little to the paper. Figure 8 on the other hand is very helpful to document lack of systemic effect of this compound at doses that are pharmacologically relevant. A dose response curve for systemic expression of these mediators would be important to help establish a therapeutic window for the compound.

We would prefer to retain panel A in former Fig. 7 (now Fig. 6) because it shows that all of the mice in all of the treatment groups lost an identical fraction of their body weights before initiation of therapy, i.e. suggesting that the severity of their pulmonary fibrosis before treatment was similar.

We have provided the additional data requested by the reviewer in new panels D, E, and F of Fig.7, where we show the concentration dependence of production of IL-6, IFN α , and TNF α in mice as a function of the administered dose of FA-TLR7-54 or nontargeted TLR7-54.

As you can determine from our responses, we have tried to comply with all of the suggestions of the three reviewers, performing multiple additional experiments to provide the additional requested data. We hope that the manuscript is now acceptable for publication in EMBO Molecular Medicine. If further changes are deemed necessary for acceptance, we would be happy to consider them.

References:

Cherfils-Vicini J, Platonova S, Gillard M, Laurans L, Validire P, Caliandro R, Magdeleinat P, Mami-Chouaib F, Dieu-Nosjean M-C, Fridman W-H et al (2010) Triggering of TLR7 and TLR8 expressed by human lung cancer cells induces cell survival and chemoresistance. JCI Insight 120: 1285-1297

Millar AB (2006) IL-10: Another therapeutic target in idiopathic pulmonary fibrosis? Thorax 61: 835-836

Misharin AV, Morales-Nebreda L, Reyfman PA, Cuda CM, Walter JM, McQuattie-Pimentel AC, Chen C-I, Anekalla KR, Joshi N, Williams KJ (2017) Monocyte-derived alveolar macrophages drive lung fibrosis and persist in the lung over the life span. J Exp Med 214: 2387-2404

Murray PJ (2017) Macrophage Polarization. Annu Rev Physiol 79: 541-566

26th May 2020

Dear Dr. Low,

Thank you for the submission of your revised manuscript to EMBO Molecular Medicine. We have now received the enclosed reports from the referees that were asked to re-assess it. As you will see the reviewers are now globally supportive and I am pleased to inform you that we will be able to accept your manuscript pending the following final amendments:

***** Reviewer's comments *****

Referee #1 (Comments on Novelty/Model System for Author):

Bleomycin is not relevant model to represent idiopathic pulmonary fibrosis (IPF). This manuscript still in my view doesn't demonstrate any relevance to human disease either by taking tissues or cells from IPF patient lungs/ BAL fluid. This hampers the medical impact.

Referee #2 (Comments on Novelty/Model System for Author):

I would have liked to see work in the TLR7-/- mouse with the drug, but given the pandemic, I can accept the work they did in cells with TLR7 or not.

Referee #2 (Remarks for Author):

The revisions have satisfied my major questions regarding the macrophage populations which are targeted and the requirement of TLR7 for the drug effect.

Referee #3 (Comments on Novelty/Model System for Author):

This MS describes a novel approach to treatment of pulmonary fibrosis, an otherwise poorly treated disease. This is an initial description of the approach using an animal model which shows that a rational modification of a known approach reduces toxicity to an acceptable level while also establishing efficacy in vivo.

Referee #3 (Remarks for Author):

This MS is a response to reviewers comments. While I still have some reservations about the phrasing and explanation of some of the results, the actual data stands on its own and is now suitable for publication.

***** Reviewer's comments *****

Referee #1 (Comments on Novelty/Model System for Author):

Bleomycin is not relevant model to represent idiopathic pulmonary fibrosis (IPF). This manuscript still in my view doesn't demonstrate any relevance to human disease either by taking tissues or cells from IPF patient lungs/ BAL fluid. This hampers the medical impact.

We thank the reviewer for pointing out the limitations of current animal models of human IPF disease. Unfortunately, there is currently no animal model that accurately replicates all aspects of human IPF. In fact, the characteristics of human IPF differ among patients, so that one human patient would also not serve as a good model for the next human patient. However, according to the Official American Thoracic Society Workshop Report, bleomycin model is still considered as “the best-characterized animal model available for preclinical testing”. We have added a sentence to page 8 that addresses the relevance of the bleomycin model to human IPF and have supported this statement with citation of two prominent references.

Referee #2 (Comments on Novelty/Model System for Author):

I would have liked to see work in the TLR7^{-/-} mouse with the drug, but given the pandemic, I can accept the work they did in cells with TLR7 or not.

Referee #2 (Remarks for Author):

The revisions have satisfied my major questions regarding the macrophage populations which are targeted and the requirement of TLR7 for the drug effect.

Referee #3 (Comments on Novelty/Model System for Author):

This MS describes a novel approach to treatment of pulmonary fibrosis, an otherwise poorly treated disease. This is an initial description of the approach using an animal model which shows that a rational modification of a known approach reduces toxicity to an acceptable level while also establishing efficacy in vivo.

We thank the reviewer's generous comments on the novelty and significance of our manuscript.

Referee #3 (Remarks for Author):

This MS is a response to reviewers' comments. While I still have some reservations about the phrasing and explanation of some of the results, the actual data stands on its own and is now suitable for publication.

We thank the reviewer's suggestions on our wording and his/her agreement with the publication.

The authors performed the requested changes.

Corresponding Author Name: Philip S. Low
Journal Submitted to: EMBO Molecular Medicine
Manuscript Number: EMM-2020-12034